# Flexible control of motor units: is the multidimensionality of motor unit manifolds a sufficient condition?

François Dernoncourt[1], Simon Avrillon[1,2] [iD], Tijn Logtens[1] [iD], Thomas Cattagni[3] [iD], Dario Farina[2] [iD] and François Hug[1,4] [iD]

[1] *Université Côte d'Azur, LAMHESS, Nice, France*
[2] *Department of Bioengineering, Faculty of Engineering, Imperial College London, London, UK*
[3] *Nantes Université, Laboratory 'Movement, Interactions, Performance' (UR 4334), Nantes, France*
[4] *The University of Queensland, School of Biomedical Sciences, Brisbane, Queensland, Australia*

Handling Editors: Richard Carson & Mathew Piasecki

The peer review history is available in the Supporting information section of this article (https://doi.org/10.1113/JP287857#support-information-section)

*The Journal of Physiology*

This article was first published as a preprint. Dernoncourt F, Avrillon S, Logtens T, Cattagni T, Farina D, Hug F. 2024. Flexible control of motor units: Is the multidimensionality of motor unit manifolds a sufficient condition? bioRxiv. https://doi.org/10.1101/2024.07.23.604408

**Abstract figure legend** We recorded large populations of motor units from the vastus lateralis (VL) and gastrocnemius medialis (GM). Using a linear dimensionality reduction approach, we observed that GM motor unit activity was effectively captured by a single latent factor, defining a unidimensional manifold. In contrast, VL motor units were better represented by three latent factors, defining a multidimensional manifold. We then evaluated the flexibility of motor unit control during sinusoidal contractions with torque feedback (torque control) and during online control tasks with visual feedback on firing rates (firing rate control). Flexibility was low regardless of the muscle. We propose that spinal circuits can shape supraspinal drive to generate multidimensional manifolds without necessarily providing additional capacity for volitional control. pps, pulses per second.

**Abstract** Understanding flexibility in the neural control of movement requires identifying the distribution of common inputs to the motor units. In this study, we identified large samples of motor units from two lower limb muscles: the vastus lateralis (VL; up to 60 motor units per participant) and the gastrocnemius medialis (GM; up to 67 motor units per participant). First, we applied a linear dimensionality reduction method to assess the dimensionality of the manifolds underlying the motor unit activity. We subsequently investigated the flexibility in motor unit control under two conditions: sinusoidal contractions with torque feedback, and online control with visual feedback on motor unit firing rates. Overall, we found that the activity of GM motor units was effectively captured by a single latent factor defining a unidimensional manifold, whereas the VL motor units were better represented by three latent factors defining a multidimensional manifold. Despite this difference in dimensionality, the recruitment of motor units in the two muscles exhibited similarly low levels of flexibility. Using a spiking network model, we tested the hypothesis that dimensionality derived from factorization does not solely represent descending cortical commands but is also influenced by spinal circuitry. We demonstrated that a heterogeneous distribution of inputs to motor units, or specific configurations of recurrent inhibitory circuits, could produce a multidimensional manifold. This study clarifies an important debated issue, demonstrating that while motor unit firings of a non-compartmentalized muscle can lie in a multidimensional manifold, the CNS may still have limited capacity for flexible control of these units.

(Received 11 October 2024; accepted after revision 27 January 2025; first published online 17 February 2025)

**Corresponding author** F. Hug: Université Côte d'Azur, LAMHESS, Nice, France. Email: Francois.hug@univ-cotedazur.fr

## Key points

- To generate movement, the CNS distributes both excitatory and inhibitory inputs to the motor units.
- The level of flexibility in the neural control of these motor units remains a topic of debate with significant implications for identifying the smallest unit of movement control.
- By combining experimental data and *in silico* models, we demonstrated that the activity of a large sample of motor units from a single muscle can be represented by a multidimensional linear manifold; however, these units show very limited flexibility in their recruitment.
- The dimensionality of the linear manifold may not directly reflect the dimensionality of descending inputs but could instead relate to the organization of local spinal circuits.

**François Dernoncourt** is a PhD candidate in Human Movement Science at Université Côte d'Azur, France. He holds a master's degree in Sports Sciences from Nantes Université, France. His PhD research explores the neural control of movement, with a focus on characterizing the smallest unit of movement control in the nervous system.

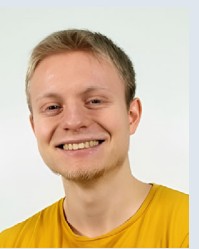

## Introduction

To generate movement, the central nervous system (CNS distributes excitatory and inhibitory inputs to the spinal motor neurons. Observations of correlated or synchronized activity among motor units have led to the assumption that a substantial portion of the net excitatory input is shared across different motor units from the same pool (de Luca et al., 1982; Negro et al., 2009; Schmied et al., 1994). The combination of common inputs with the size principle (Henneman, 1957) theoretically imposes rigid control on motor units, which is characterized by a consistent size-ordered recruitment pattern. Although this control strategy is computationally efficient at simplifying the neural control of motor units, it has been challenged by the concept of a more flexible control, in which motor unit recruitment may deviate from the size principle (Desmedt & Godaux, 1981; Formento et al., 2021; Herrmann & Flanders, 1998; Marshall et al., 2022; Ter Haar Romeny et al., 1982). Understanding the flexibility in motor unit recruitment requires the identification of the distribution of common inputs (Bawa et al., 2014; Hug et al., 2023).

One approach for identifying the structure of common inputs to motor units is to apply linear dimensionality reduction methods to a series of firing times for many units. This approach can capture common patterns in the firing activities of motor units within low-dimensional manifolds shaped by the dynamics of latent factors. Using this approach, the firing patterns of the motor units of certain muscles have been projected onto multidimensional manifolds (Del Vecchio et al., 2023; Madarshahian & Latash, 2021; Ricotta et al., 2023). Such a multidimensional manifold of motor unit firings can be interpreted as the presence of multiple common inputs (Hug et al., 2023). This observation may align with the concept of task-dependent behaviour of motor units, and thus it may support flexibility in motor unit control, possibly of cortical origin (Marshall et al., 2022). Although this interpretation is plausible, the neural pathways responsible for increasing the dimensionality of motor unit manifolds remain unclear. Motor neurons receive numerous excitatory and inhibitory inputs from both cortical and subcortical sources. As such, a multidimensional manifold could emerge from a single common input of a cortical origin combined with additional common inputs from subcortical sources. This interplay between inputs from different origins could explain the contradictory results among studies. For example, Bräcklein et al. (2022) observed that participants could not independently control the motor units of the tibialis anterior muscle, whereas Ricotta et al. (2023) showed that the motor unit firings in this muscle lie in a multidimensional linear manifold.

In the present study, we hypothesized that common cortical projections to motor neurons are not necessarily the sole inputs that determine the dimensionality of the motor unit manifolds. Other types of inputs, not under direct cortical control, may also shape a manifold's dimensionality. For example, the activity of Renshaw cells, which are activated by recurrent collaterals of homonymous and heteronymous motor neurons (Adam et al., 1978; Eccles et al., 1961), can contribute to decorrelating motor unit activity (Maltenfort et al., 1998). Therefore, this circuit can contribute to increasing the dimensionality of the firing manifold, without increasing the flexibility of volitional motor unit control. We assume that the flexible control of motor units is determined primarily by the number of cortical inputs, rather than by the total number of inputs shaping the dimensionality of the manifold. It is possible that, in some muscles, a multidimensional linear manifold underlies a single cortical input, resulting in a rigid control of motor units.

To test this hypothesis, we recorded large samples of motor units to unravel the low-dimensional structures of their inputs. We selected two lower limb muscles based on previous studies reporting different levels of correlated activity of their motor units, with the vastus lateralis (VL) exhibiting less correlated activity (Avrillon et al., 2021) compared to the gastrocnemius medialis (GM) (Hug, del Vecchio et al., 2021). We investigated whether the expected difference in manifold dimensionality between the two muscles reflected different levels of flexibility in their motor unit activity and, therefore, different neural constraints on motor unit control. First, we assessed the dimensionality of the manifolds underlying motor unit activity during torque-matched isometric contractions. We then investigated the flexibility in motor unit control under two conditions: sinusoidal contractions with torque feedback and online control with visual feedback on motor unit firing rates. Despite the differences in linear manifold dimensionality between the muscles, both exhibited similar and very limited flexibility in motor unit control. Finally, we developed a simulation model that demonstrated that the multidimensionality of the linear manifold in the VL was compatible with the presence of a heterogeneous distribution of inputs to motor units, or specific configurations of recurrent inhibitory circuits, even with a single descending command. These results suggest that the VL motor units receive common inputs from various sources, not all of which are under volitional control. This study clarifies an important debated issue in motor unit control, by showing that motor unit firings can lie in a multidimensional manifold; however, it may still be impossible for the CNS to flexibly control these motor units.

## Methods

### Participants and ethical approval

The first experiment involved a series of isometric torque-matched knee extension and plantarflexion tasks. Nine physically active males (age: 26.7 ± 7.6 years, height: 179.7 ± 7.9 cm, body mass: 76.6 ± 11.6 kg) performed the knee extension tasks while seven active males (age: 29.2 ± 7.2 years, height: 176.8 ± 2.8 cm, body mass: 77.8 ± 10.1 kg) performed the plantarflexion tasks. Two participants performed both the knee extension and plantarflexion tasks in two separate sessions. Owing to the low number of identified motor units (typically fewer than 10), two participants were excluded from the analysis of the knee extension task, and one participant was excluded from the plantarflexion task.

The second experiment involved isometric knee extension or plantarflexion tasks, during which the participants controlled the activity of pairs of motor units using online visual feedback on their firing rate. Six physically active males performed the knee extension tasks (age: 29.2 ± 7.9 years, height: 175.2 ± 3.1 cm, body mass: 73.8 ± 8.6 kg) and the plantarflexion tasks (age: 29.4 ± 7.7 years, height: 176.2 ± 3.9 cm, body mass: 74.8 ± 8.4 kg). Five of these participants completed both tasks across two separate sessions.

Participants had no history of lower leg pain with limited function that required time off work or physical activity, or a consultation with a health practitioner in the previous 6 months. The procedures were approved by the ethics committee (CPP Ouest III; 23.00453.000166), and were conducted in accordance with the *Declaration of Helsinki*, except for registration in a database. The participants were fully informed of any risks or discomfort associated with the procedures before providing written informed consent.

### Experimental design

For the knee extension tasks, the participants sat on a dynamometer (S2P, Ljubljana, Slovenia) with their hips and right leg each flexed at 80° (0° being the neutral position). For the plantarflexion tasks, the participants sat on a dynamometer (Biodex System 3 Pro, Biodex Medical, USA) with their hips flexed at 80° and their right legs fully extended. The foot was placed perpendicular to the shank. During both tasks, inextensible straps were tightened to immobilize the torso, pelvis and thighs on the test side. These isometric tasks were used in the two experiments. In the first experiment, the participants matched the visual torque feedback. In the second experiment, they aimed to modulate the activity of two motor units using online visual feedback displaying their firing rates. In both experiments, sessions began with a standardized warm up,

followed by three maximal voluntary contractions (MVC) for 3 s, with 60 s of rest in between. Peak MVC torque was considered as the maximal value obtained from a moving average window of 250 ms.

**Experiment 1 (torque-matched contractions).** Participants performed four trapezoid isometric contractions, each consisting of a 5 s linear ramp-up, a 20 s plateau at 20% MVC and a 5 s linear ramp-down, with 30 s of rest in between. Participants then performed sinusoidal contractions, producing torque oscillations at frequencies of either 0.25, 1 and 3 Hz. Each sinusoidal contraction consisted of a 5 s linear ramp-up to 20% MVC, followed by a 10 s plateau at 20% MVC, and 20 s of sinusoidal oscillations between 15% and 25% MVC. Each sinusoidal contraction was performed three times, with at least 60 s of rest in between. The order of the frequency conditions was intentionally not randomized to allow the participants to progressively familiarize themselves with faster oscillations. For the 3 Hz sinusoidal condition, auditory feedback was provided in addition to visual feedback using a metronome.

**Experiment 2 (volitional control of motor units).** Participants performed a trapezoid isometric contraction consisting of a 5 s linear ramp-up, a 20 s plateau at either 10% (knee extension) or 20% MVC (plantarflexion), and a 5 s linear ramp-down (referred to as 'baseline' contraction). The lower torque level for the knee extension tasks was chosen to minimize fatigue, which develops faster during knee extension than plantarflexion (Rossato et al., 2024a). This baseline contraction was used to identify motor unit separation vectors (filters) offline, which were then reapplied in real-time to display the motor unit firing rate as visual feedback (Rossato, Hug et al., 2024). After a 15–30 min rest period during which automatic offline decomposition of the EMG signals and manual editing of the motor units were performed, the participants performed four 20 s submaximal isometric contractions, with 1 min of rest in between. Only the torque feedback was provided to the participants during these contractions (referred to as 'reference' contractions).

The rest of the experimental session consisted of a series of 10 s isometric contractions during which the participants were instructed to dissociate the activity of two motor units while maintaining constant knee extension (10% MVC) or plantarflexion (20% MVC) torque constant. Specifically, the activity of the two motor units was depicted in 2D space, where the *x*- and *y*-axes represented the firing rates of motor units #1 and #2, respectively. The scale of each axis ranged from 0 to twice the average firing rate of the corresponding motor units measured during baseline contraction. This scaling ensured that the cursor was not biased towards any of the

axes. Two target triangles were added to the 2D space, each with one corner on the coordinate origin and two sides as one axis (*x* or *y*), and a line forming an angle of 30° along this axis. In addition, real-time feedback on torque output was provided to the participants via a vertical gauge, which they were required to maintain within two lines representing ±5% of the target torque.

For a given pair of motor units, the participants performed two sets of ten 10 s contractions, with 10 s of rest between each contraction. The participants aimed to reach Targets 2 and 1 during the first and second sets, respectively. To move the cursor towards these targets, the participants were required to differentially alter the firing rates of the two motor units (i.e. an increase in one unit with no change or a decrease in the other unit). This procedure was repeated with two to four different pairs of motor units, depending on the participant. Each pair comprised a motor unit from each of two EMG grids (proximal and distal; see below), providing participants with information regarding the location of the motor units they were asked to control independently.

## Decoding of motor unit firing activity

**High-density surface EMG recordings.** EMG signals were recorded from either the GM (plantarflexion) or the VL, rectus femoris (RF) and vastus medialis (VM) (knee extension). For the first experiment, four adhesive grids of 64 electrodes (total of 256 electrodes) were placed over the VL (inter-electrode distance: 8 mm; GR08MM1305, OT Bioelettronica, Turin, Italy) or GM (inter-electrode distance: 4 mm; GR04MM1305, OT Bioelettronica). One grid each of 64 electrodes was placed over the RF and VM muscles (inter-electrode distance: 8 mm; GR08MM1305, OT Bioelettronica). For the second experiment, two adhesive grids of 64 electrodes (a total of 128 electrodes) were placed over the proximal and distal region of the VL (inter-electrode distance: 8 mm; GR08MM1305, OT Bioelettronica) or GM muscle (inter-electrode distance: 4 mm; GR04MM1305, OT Bioelettronica). Overall, this setup was selected to maximize the number of decomposed motor units (Caillet et al., 2023).

Prior to electrode application, the skin was shaved and cleaned with an abrasive paste (Nuprep, Weaver and Company, Aurora, CO, USA). Adhesive grids were held on the skin using bi-adhesive foam layers (SpesMedica, Battipaglia, Italy). The skin–electrode contact was made by filling the cavities of the adhesive layers with conductive paste (SpesMedica, Battipaglia, Italy). A reference electrode (5 × 5 cm; Kendall Medi-Trace; Covidien, Dublin, Ireland) was placed over the right patella (knee extension tasks) or right tibia (ankle plantarflexion tasks), and a strap electrode dampened with water (ground electrode) was placed around the left ankle. The EMG

signals were recorded in a monopolar montage, band-pass filtered (10–500 Hz), and digitized at a sampling rate of 2048 Hz using a multichannel acquisition system (EMG-Quattrocento; 400-channel EMG amplifier; OT Biolelettronica).

**Offline EMG decomposition and motor unit tracking.** The EMG signals were decomposed using convolutive blind source separation (Holobar & Zazula, 2007; Negro, Muceli, et al., 2016), and implemented in the MUedit software (Avrillon et al., 2024). Prior to automatic decomposition, the channels were visually inspected, and those with low signal-to-noise ratio or artifacts were discarded. The remaining EMG signals were extended and whitened. A fast independent component analysis was performed to retrieve the motor unit spike trains mixed within the EMG signals. Specifically, a separation vector was optimized for each motor unit using a fixed-point algorithm to obtain a sparse motor unit pulse train. Firing times were identified from these pulse trains using a peak detection function and a *k*-mean classification that separated the high peaks (the firing times) from the low peaks (noise). The distance between the high and low peaks was quantified using the silhouette value, and only the pulse trains with a high distance between the high and low peaks (threshold: 0.9) were retained for subsequent analyses. The resulting spike trains were then edited manually. Manual editing is an iterative process that involves removing detected peaks that result in erroneous firing rates (outliers), and adding missed firing times that are clearly distinguishable from the noise. The motor unit pulse trains are ultimately recalculated with updated separation vectors and accepted by the operator once all the putative firing times are selected. This procedure was performed according to the guidelines described by Del Vecchio et al. (2020), and was demonstrated to be highly reliable across operators (Hug, Avrillon et al., 2021).

The decomposition was performed independently on each electrode grid. Given that the same motor units could be detected across multiple grids covering the same muscle, duplicate units were identified through analysis of their pulse trains. The pulse trains of pairs of motor units were first aligned using a cross-correlation function to account for a potential delay owing to the propagation time of the action potentials along the fibres. Firing times that occurred within 0.5 ms intervals were considered as common between motor units; and motor units that shared more than 30% of their firing times were considered as duplicates (Holobar et al., 2010). When duplicated motor units were identified, only the motor unit with the lowest coefficient of variation of its inter-spike intervals was retained.

For the first experiment, the decomposition procedure described above was performed on each task separately,

namely each trapezoidal reference contraction and each sinusoidal contraction. For the trapezoidal reference contractions, results are reported for the motor units tracked across all four contractions. For the sinusoidal contractions, results are reported for both the untracked and tracked motor units across all three frequency conditions. To track motor units, the separation vectors of the motor unit identified in one contraction were applied to the extended and whitened EMG signals of the other contractions (Frančič & Holobar, 2021). The resulting spike trains were then compared with those obtained after manual editing. A motor unit was tracked if at least 30% of its spikes were shared. This process was applied to each potential pair of contractions. This automatic process was complemented with a visual inspection of the 2D spatial distribution of motor unit action potentials across the 256 electrodes to confirm the accuracy of the motor unit tracking. Notably, motor units were not tracked between the reference and sinusoidal contractions due to a large decrease in motor unit yield. The difficulty in tracking motor units across contractions with differing mechanical constraints – and thus different activation levels (e.g. trapezoidal *vs.* sinusoidal tasks) – probably arises from technical limitations of the decomposition approach. Specifically, tracking smaller motor units becomes challenging during contractions performed at higher intensities than those at which they were initially identified, due to the reduced signal-to-noise ratio when larger motor units are recruited. However, we verified that outcomes remained consistent even with the smaller sample of tracked units (results not reported here).

**Real-time EMG decomposition.** To provide online feedback on the motor unit activity in the second experiment, the EMG signals were decomposed in real time using open-source software developed and validated by our team (Rossato, Hug et al., 2024). First, EMG signals were recorded during a submaximal trapezoid contraction and decomposed offline to identify the motor unit separation vectors and the centroids of the classes 'spikes' and 'noise' were calculated with a peak detection function and $k$-mean classification. As the accuracy of the online decomposition can be improved by manually editing the offline decomposition results (Rossato et al., 2024a), manual editing was performed prior to the real-time component of the experiment.

For real-time EMG decomposition, motor unit separation vectors were applied to the incoming segments of 125 ms of EMG signals. Peaks closer to the centroid of the 'spikes' class than that of the 'noise' class were considered as firing times. Besides the 125 ms incompressible delay, the computational time for visual feedback was typically less than 15 ms (Rossato et al., 2024a). We calculated the firing rate

of individual motor units as the sum of spikes over eight consecutive windows of 125 ms. This approach introduced a 500 ms delay in estimating the firing rate, but allowed for smoother and more controllable visual feedback than using instantaneous firing rates (Rossato et al., 2024a).

### Data analysis

**Factor analysis.** We conducted a factor analysis on the smoothed firing rates of the motor units identified from the GM or VL muscle during the reference contractions (experiment 1) using the 'factoran()' function in Matlab (R2023b, Mathworks, Natick, MA, USA). First, the binary spike trains were convolved with a 400 ms kernel Hanning window without a phase shift (Del Vecchio et al., 2023). This corresponds to a 2.5 Hz low-pass filter, which retains the information relevant to force production (Farina et al., 2014; Negro, Yavuz, et al., 2016). Subsequently, the four 20 s force plateaus were concatenated, and only motor units continuously firing, without any pause between two firings longer than 400 ms, were included in the analysis. Finally, the smoothed firing rates were detrended (De Luca & Erim, 2002).

Factor analysis allows for factor rotation, which maximizes the overall correlation between the original variables and latent factors, without requiring the latent factors to be orthogonal. We used *promax* rotation, which aims to simplify the structure of the loading matrix by maximizing the weight of one factor, while minimizing the weights of other factors for each variable. To determine the appropriate number of latent factors, we adopted the method proposed by Cheung et al. (2009). This approach involves selecting the minimal number of latent factors beyond which additional factors do not significantly improve the model's capability to reconstruct the data over what would be expected from fitting random noise. First, we iterated the factor analysis by varying the number of latent factors between one and ten. For each number of latent factors, the reconstruction accuracy was calculated as the $R^2$ value between the reconstructed and the original data. We also generated a surrogate dataset through a random permutation of the motor unit spike trains. The surrogate dataset had the same number of spikes and the same mean firing rate as the original dataset. We performed factor analysis on this surrogate dataset, and the $R^2$ value was calculated for each latent factor from one to ten. The slope of the $R^2$ curve for the surrogate data was determined by fitting a straight line using least squares regression. The critical number of factors was determined as the maximum count at which the slope of the $R^2$ curve of the original dataset exceeded that of the surrogate data. Beyond this point, additional factors were considered to capture noise. Notably, we also

explored alternative methods for selecting the number of latent factors. One method involved the selection of the number of factors beyond which the increase in $R^2$ was lower than 5% (Clark et al., 2010). Another method involved selecting the number of factors at which the $R^2$ curve plateaued, as indicated by a mean square error of the linear fit being less than $10^{-3}$ (Cheung et al., 2005).

We also examined whether factor analysis identified latent factors based on the intrinsic properties of motor units. For each participant, we projected the motor units onto the latent factors, assigning each motor unit a position vector in the factor space. The coordinates of this vector reflected the correlation between the unit's smoothed discharge rates and each factor. Motor units were subsequently clustered based on their dominant factor, identified as the factor with the highest magnitude. This approach allowed us to evaluate whether motor unit clustering was associated with firing rate and/or recruitment threshold.

**Dispersion and displacement.** We calculated two metrics proposed by Marshall et al. (2022) to determine whether the firing rates of motor unit populations deviated from the expected behaviour of motor units exclusively receiving a 1D common input: motor unit displacement and dispersion (Fig. 2). The firing rates of a population of motor units receiving a 1D common input should constantly vary along a 1D monotonic manifold in the same direction as that of the common input. If the firing rate of one motor unit increases, all other firing rates should either increase, remain at zero because of a later recruitment threshold, or plateau at the same value because of rate saturation. The motor unit displacement quantifies the difference in firing rates between two motor units when the firing rates of the two units vary in opposite directions. For this, we first computed the maximum non-negative changes in the firing rates between two timestamps, $t$ and $t'$, for a population of $n$ motor units.

$$\Delta r(t, t\prime)$$
$$= \max(r1, t - r1, t\prime, \ r2, t - r2, t\prime, \ \ldots, \ rn, t - rn, t\prime)$$

If all the motor unit firing rates vary in the same direction, then either $\Delta(t, t')$ or $\Delta r(t', t)$ should be equal to 0. The displacement ($t$) was considered as the minimum value between $\Delta(t, t')$ and $\Delta r(t', t)$. Of note, we quantified the displacement values while iteratively adding time lags $\tau$ and $\tau$' of up to 25 ms to $t$ and $t'$, respectively, to account for differences in conduction velocities between motor neurons. Therefore, we maintained the minimal displacement value between $\Delta(t + \tau, t' + \tau')$ and $\Delta r(t' + \tau', t + \tau)$. Finally, we considered the displacement for a population of motor units as the maximum value of all the time stamps $t$.

Dispersion quantifies the maximal deviation of variations in firing rates from a 1D manifold at the population level. Thus, the summed firing rates of a population of motor units receiving a 1D common input, equivalent to its L1-norm $\|r\|$, should always be associated with the same population state, namely the distribution of firing rates within the population. Any change in L1-norm within a population of motor units for a given norm value l would mean that at least two motor units have different firing rates for the same summed firing rate. The dispersion quantifies the maximal difference between L1-norms for a given norm value λ:

$$(\lambda) = \max([[rt1 - rt2]])$$

We further quantified the dispersion values while iteratively adding time lag vectors $\tau$ and $\tau$' of up to 25 ms to the population of the motor units.

Note that we calculated the displacement and dispersion metrics for all possible pairs of motor units. To facilitate the comparison of motor unit flexibility between muscles and motor tasks with different mean firing rates, we normalized the displacement and dispersion values by dividing the values by the sum of the maximum firing rates of the motor units in each pair.

**Distance between motor units.** To approximate the distance between motor units within the muscle, motor unit action potential waveforms were calculated by spike-trigger-averaging the differential EMG signals over 50 ms windows for each grid of electrodes. The peak-to-peak amplitude of each waveform was calculated, resulting in a 12 × 5 array of amplitude values for each grid and motor unit. To mitigate the risk of encountering aberrant values due to the presence of artifacts in the EMG signal, we applied a 3 × 3 box blur kernel to each of the 12 × 5 arrays. The electrode with the highest amplitude value across the four grids was considered as the approximate location of the motor unit. These positions were then used to calculate the distances between motor units for all possible pairs, assuming an 8 mm inter-electrode distance and a 24 mm inter-grid distance.

**Recruitment threshold.** Recruitment time was defined as the time of the first firing of a series of three firings occurring within 1 s. For each motor unit, the recruitment threshold was calculated as the average torque (expressed as a percentage of MVC) recorded at the recruitment times across all contractions.

### Spiking neural network model

We used the Brian2 package implemented in Python (Stimberg et al., 2019) to simulate the behaviour

of motor neuron populations under seven different scenarios: (i) a single common excitatory input, (ii) three excitatory inputs distributed homogeneously or (iii) heterogeneously, (iv) three inhibitory inputs distributed homogeneously or (v) heterogeneously, (vi) homonymous recurrent inhibition, and (vii) both homonymous and heteronymous recurrent inhibition. The code and the list of parameters are publicly accessible at: https://github.com/FrancoisDernoncourt/Motor_unit_flexibility

**Motor neuron model.** We used a conductance-based leaky integrate-and-fire model where each neuron was defined by a resting membrane potential and reset voltage $E_{rest}$ of 0 mV, and a firing threshold $V_{thresh}$ of 10 mV. The voltage change per timestep, $\frac{dv}{dt}$, was calculated based on the leak current $I_{leak}$ (in nA), excitatory current $I_{excit}$ (in nA), inhibitory current $I_{inhib}$ (in nA) and the membrane capacitance $C$ (in F), using the following equation:

$$\frac{dv}{dt} = \frac{-(I_{leak} + I_{excit} + I_{inhib})}{C}$$

The current values were calculated based on the specific conductance values ($g$) and equilibrium potential ($E$), as follows:

$$I_{leak} = g_{leak} \times (v - E_{rest})$$

$$I_{excit} = \max(g_{excit} \times (v - E_{excit}) + I_{rheobase}, 0)$$

$$I_{inhib} = g_{inhib} \times (v - E_{inhib})$$

An offset $I_{rheobase}$ was applied to the excitatory current, representing the minimum current required for the excitatory conductance to be non-zero. $E_{excit}$ was set to 25 mV and $E_{inhib}$ to −15 mV. These values ensured that excitatory and inhibitory inputs produced voltage changes of equal magnitude when the membrane potential was halfway between $E_{rest}$ and $V_{thresh}$.

The conductance values (in mS) were defined as:

$$g_{leak} = \frac{1}{R_i}$$

$$g_{excit} = \gamma_{excit}(t, i) \times S_i$$

$$g_{inhib} = \gamma_{inhib}(t, i) \times S_i$$

where $R_i$ (in ohms) represents the membrane resistance of motor neuron $i$, $\gamma_{excit}(t, i)$ and $\gamma_{inhib}(t, i)$ are the time-dependent excitatory and inhibitory synaptic inputs delivered to motor neuron $i$ at each timestep $t$, and $S_i$ is a scaling factor representing the responsiveness of the membrane potential to a given input. $S_i$ values were calculated as the normalized resistance of motor neurons, with $S_i$ of the smallest motor neuron set to 1. After firing, motor neurons underwent a refractory period $T_{refractory}$ during which the membrane potential was clamped to 0 mV.

Each motor pool consisted of 300 simulated motor neurons. Each motor neuron $i$ within a pool was assigned a specific soma diameter $D_i$ (ranging from 50 [$D_{min}$] to 100 [$D_{max}$] μm) based on the following quadratic function:

$$D_i = D_{min} + \left(\frac{i}{N}\right)^2 \times (D_{max} - D_{min})$$

This distribution ensured a higher representation of low-threshold motor neurons (Duchateau & Enoka, 2022). The relationship between the soma diameter and $R$, $C$, $I_{rheobase}$ and $T_{refractory}$ was determined using the equations provided by Caillet et al. (2022) (for further details, please refer to the code).

**Input to the simulated motor neurons.** The simulated motor neurons received a mixture of inputs in proportions similar to those proposed by Farina and Negro (2015): 25% excitatory and/or inhibitory signals (0–2.5 Hz bandwidth), and 75% independent noise (0–50 Hz bandwidth). These input signals had a mean of 0 mS and a standard deviation of 0.03 mS, and they were added to a baseline excitatory input. The baseline excitatory input ranged from 0.5 mS in scenarios with only excitatory inputs (scenarios i, ii and iii) to 0.78 mS in the scenario involving recurrent inhibition from homonymous and heteronymous pools (scenario vii). When present, the baseline inhibitory input was set to 0.12 mS.

The common inhibitory inputs and the independent inputs were modelled as low-pass-filtered Gaussian noise with cutoff frequencies of 2.5 and 50 Hz, respectively. The excitatory common inputs were defined as the standardized first principal component(s) derived from the smoothed firing rates recorded during the reference contraction, implying that these excitatory inputs were orthogonal. However, in the scenario involving recurrent inhibition across heteronymous motor pools, we applied an arbitrary pairwise correlation coefficient of ∼0.7, ensuring that these (synergistic) pools received correlated inputs, consistent with the high proportion of shared inputs observed among the Vastii (Laine et al., 2015, p. 20; Avrillon et al., 2021).

When inputs were homogeneously distributed, each motor neuron received an equal proportion of each input. In scenarios with a heterogeneous distribution, input weights were randomly sampled from a uniform distribution ranging from 0 to 1 and then processed using a Softmax function with a temperature parameter of 0.1. This procedure ensured that the weights were summed to one while preserving a heterogeneous distribution of inputs.

**Recurrent inhibition simulation.** In scenarios vi and vii, we incorporated Renshaw cells into the model as simple

integrate-and-fire neurons. The membrane voltage of each Renshaw cell evolved according to:

$$\frac{\mathrm{d}v}{\mathrm{d}t} = \frac{I_{\mathrm{Renshaw}} - v}{\tau}$$

where $I_{\mathrm{Renshaw}}$ is the baseline independent input (in mV), and $\tau$ is the time constant set to 8 ms (Maltenfort et al., 1998; Williams & Baker, 2009). Each Renshaw cell had a resting membrane potential and reset voltage of 0 mV, a firing threshold of 10 mV and a refractory period of 10 ms.

Each Renshaw cell received independent input modelled as zero-mean, 50 Hz low-pass-filtered Gaussian noise with a standard deviation of 3 mV. The excitatory post-synaptic potential from a motor neuron to its target Renshaw cells was 1.5 mV, and the inhibitory post-synaptic potential from a Renshaw cell to its target motor neurons was 0.002 mS. There was a 5 ms synaptic delay in both directions (from motor neuron to Renshaw cell and from Renshaw cell to motor neuron).

Scenario vi represented homonymous recurrent inhibition with a uniform probability of connectivity between Renshaw cells and motor neurons. Specifically, the model included 60 Renshaw cells serving a single pool of motor neurons. Each Renshaw cell received input from a random subset of 17% of the motor neurons, while each motor neuron received input from ∼40% of the Renshaw cells. This connectivity pattern was designed to align with previous simulations (Williams & Baker, 2009).

Scenario vii included both homonymous and heteronymous recurrent inhibition. We simulated three pools of motor neurons, each associated with a dedicated set of 60 Renshaw cells. Within each pool, each Renshaw cell received input from a random subset of 50% of the homonymous motor neurons, while no input was received from heteronymous motor neurons. Conversely, each motor neuron received input from 60% of the Renshaw cells from its own pool. To include heteronymous connections, each motor neuron pool was divided into three subgroups. One subgroup received additional inhibitory input from 60% of the Renshaw cells of one heteronymous pool, another subgroup received input from 60% of the Renshaw cells of the other heteronymous pool, and the third subgroup received no heteronymous inhibition. Importantly, all motor neurons were consistently inhibited by their homonymous Renshaw cells, and any heteronymous recurrent inhibition originated from a distinct, non-overlapping set of Renshaw cells. This configuration aligns with evidence that recurrent inhibition can be topographically organized across homonymous and synergist pools (Brownstone & Bui, 2010; McCurdy & Hamm, 1994). Of note, when considering all three pools together, the overall percentages of connections between motor neurons and Renshaw cells were similar to those used in scenario vi.

**Simulation parameters and analysis of simulated data.** Each simulation lasted 60 s. The input amplitudes, as well as the excitatory and inhibitory post-synaptic potentials of motor neurons and Renshaw cells, were manually adjusted to produce physiologically plausible firing rates (8–10.5 Hz for motor neurons and 20–25 Hz for Renshaw cells). We pre-processed and analysed the simulated motor neuron spike trains following the same procedure applied to the experimentally recorded data (see above). In each simulation, ∼120–150 motor neurons out of the 300 were continuously active and were therefore included in the factor analysis. In scenario vii, the factor analysis was performed only on the first motor neuron pool.

### Statistical analysis

All statistical analyses were conducted using the R software (R foundation for statistical computing, Vienna, Austria). The significance threshold was set to 0.05 and adjusted using the Bonferroni correction for multiple tests. Logistic regression models were used to compare the *dispersion* and *displacement* values (dependent variables) between the muscles and contraction types (predictors), with one model per dependent variable. In all models, 'Muscle' (GM–GM; VL–VL), 'Contraction type' (sinusoidal 0.25 Hz; sinusoidal 1 Hz; sinusoidal 3 Hz, all sinusoids) and the interaction between 'Muscle' and 'Contraction type' were specified as fixed factors. For each dependent variable, we compared three different model structures with random effects of increasing complexity: no random effects, one intercept per participant and one intercept per contraction type per participant. Notably, we did not use the 'Muscle' factor in the random effect because the VL and GM experiments involved different participants. In all cases, using one intercept per contraction type per participant as a random effect resulted in the lowest Akaike information criterion and Bayesian information criterion values ($P < 0.001$). The selected models thus had the following structure: '*model ≤ lmer(dependent_variable [dispersion or displacement] ∼ muscle * contraction_type + (contraction_type | participant))*'. The statistical significance of the fixed factors was tested using the 'anova()' function. *Post hoc* pairwise comparisons consisted of contrasting estimated marginal means against each other with Bonferroni-adjusted repeated *t* tests. These comparisons were performed using the 'emmeans()' function (emmeans package). When testing for the effect of contraction types conditioning on muscles, a repeated-measures design was considered.

To assess whether factor analysis tended to cluster motor units based on their intrinsic properties, we used logistic regression models with motor unit mean firing rates or recruitment thresholds as the dependent

variable and the dominant latent factor as the predictor, including participant as a random effect. The model syntax was: 'model ≤ lmer(dependent_variable [firing_rate or recruitment_threshold] ~ dominant_factor + (dominant_factor | participant))'. To determine whether the dispersion or displacement scores of motor unit pairs were associated with the distance between the units, we used linear mixed models. The syntax for these models was: 'model ≤ lmer(dependent_variable [dispersion or displacement] ~ distance + (contraction_type | participant))'.

## Results

### Experiment 1: torque-matched contractions

Participants performed either isometric knee extension or plantarflexion tasks. These tasks consisted of matching either a trapezoidal target at 20% MVC (referred to as 'reference' contractions), or a sinusoidal target oscillating between 15 and 25% MVC at three different frequencies: 0.25, 1 and 3 Hz (referred to as 'sinusoidal' contractions). EMG signals were recorded with grids of surface electrodes. A blind source-separation algorithm (Holobar & Zazula, 2007) was applied to the EMG signals to identify motor unit spike trains. Using multiple grids of electrodes per muscle optimized the spatial sampling of motor unit surface action potentials (Caillet et al., 2023), resulting in a large sample of identified motor units (VL: up to 60 motor units per participant and GM; up to 67 motor units per participant).

**Low-dimensional analysis.** We applied a linear dimensionality reduction method, a common approach to characterize the dynamics of populations of neurons (Cunningham & Yu, 2014; DePasquale et al., 2023; Gallego et al., 2017; Shenoy et al., 2013). In our study, dimensionality reduction was applied to the firing rates of the detected motor units to identify a low-dimensional latent space. Specifically, we conducted a factor analysis on the motor units that continuously fired during the plateau phase of the reference contractions (Fig. 1). This analysis included an average of $38.1 \pm 8.1$ (range: 26–48) motor units per participant for the VL, and $28.0 \pm 19.8$ (range: 9–60) motor units per participant for the GM. To determine the dimensionality of the latent space, we identified the number of latent factors beyond which adding another factor did not improve the model's ability to reconstruct the data beyond what would be expected from fitting random noise (Cheung et al., 2009). Using this approach, we identified three latent factors in the VL muscle for all participants, accounting for $55\% \pm 7\%$ (range: 43–65%) of the total variance in motor unit firing rates. When considering the GM muscle,

only one latent factor was identified in five out of six participants, accounting for $69\% \pm 11\%$ (range: 52–82%) of the total variance. For the remaining participant, two latent factors were identified, accounting for 60% of the total variance. Notably, we also used alternative methods to determine the number of latent variables (Cheung et al., 2005; Clark et al., 2010), which yielded the same results (see the Methods section). These results indicate a distinct dimensionality of the latent spaces of the VL and GM muscles, as suggested by previous work (Del Vecchio et al., 2023; Levine et al., 2023). We also examined whether the latent factor representation of VL motor units was influenced by firing rates or recruitment thresholds rather than by the dimensionality of their inputs. No significant association emerged between dominant factors and either recruitment thresholds ($P = 0.323$) or mean firing rates ($P = 0.17$), suggesting that the dimensional structure is not determined by these intrinsic properties.

**Flexibility in motor unit control.** Based on recent work reporting flexibility in recruitment during sinusoidal tasks (Marshall et al., 2022), we evaluated the motor unit recruitment during isometric contractions with sinusoidal force profiles at various frequencies. Specifically, flexibility was quantified using two metrics: *dispersion* and *displacement* (Marshall et al., 2022) (Fig. 2). The rationale for using these metrics is based on the premise that the rigid control of motor units by a single common input would constrain the trajectory of their firing rate along a 1D monotonic manifold. While small deviations from this monotonic 1D manifold would reflect the presence of synaptic noise, large deviations would suggest that the motor units receive distinct inputs, resulting in a certain level of flexibility. These two metrics are formally defined in the Methods section. Across the three contraction frequencies, this analysis included a mean of $39.2 \pm 7.6$ motor units per participant for the VL (mean number of pairs: $5723 \pm 1862$), and $27.5 \pm 16.9$ (mean number of pairs: $3854 \pm 3364$) motor units per participant for the GM. We also calculated *dispersion* and *displacement* across all three tasks concatenated, using motor units successfully tracked across all three tasks, resulting in 186 GM–GM pairs and 1239 VL–VL pairs.

In this paragraph, 'motor unit pairs' (GM–GM and VL–VL) refers to the *Muscle* effect, while the 'frequency of the sinusoidal contractions' (0.25, 1 and 3 Hz, and all sinusoids) refers to the *Contraction* effect. Although there was no significant main effect of *Muscle* on either *dispersion* ($P = 0.34$) or *displacement* ($P = 0.99$), we did observe a significant main effect of *Contraction* ($P < 0.001$ for both metrics), as well as a significant interaction between *Muscle* and *Contraction* ($P < 0.001$ for both metrics). For the sake of clarity, we present only the differences between muscles that are directly relevant to

our main objective. There was no difference between the GM and VL either for *dispersion* or for *displacement* in all conditions (all *P* > 0.10), except for the 1-Hz sinusoidal contraction, during which the GM motor units displayed higher dispersion (*P* < 0.001) and higher displacement (*P* < 0.001) than the VL motor units (Fig. 2). Further, we noted no significant correlation between the distance separating the two motor units on the skin surface (as evaluated from their action potentials across the four grids) and the degree of flexibility, as measured using *dispersion* (mean *r* value across participants/conditions:

0.03 ± 0.12, *P* = 0.20) or *displacement* (*r* = −0.01 ± 0.11, *P* = 0.42).

To determine whether the level of flexibility estimated through *dispersion* and *displacement* was significant or negligible, we calculated both metrics for pairs of motor units from the RF and VL muscles. Given that the RF muscle is biarticular, and can be activated independently of the vastus muscles (Hug et al., 2014), the *displacement* and *dispersion* values for pairs of RF–VL motor units were used to provide insights into the expected degree of flexibility of motor units that receive different inputs.

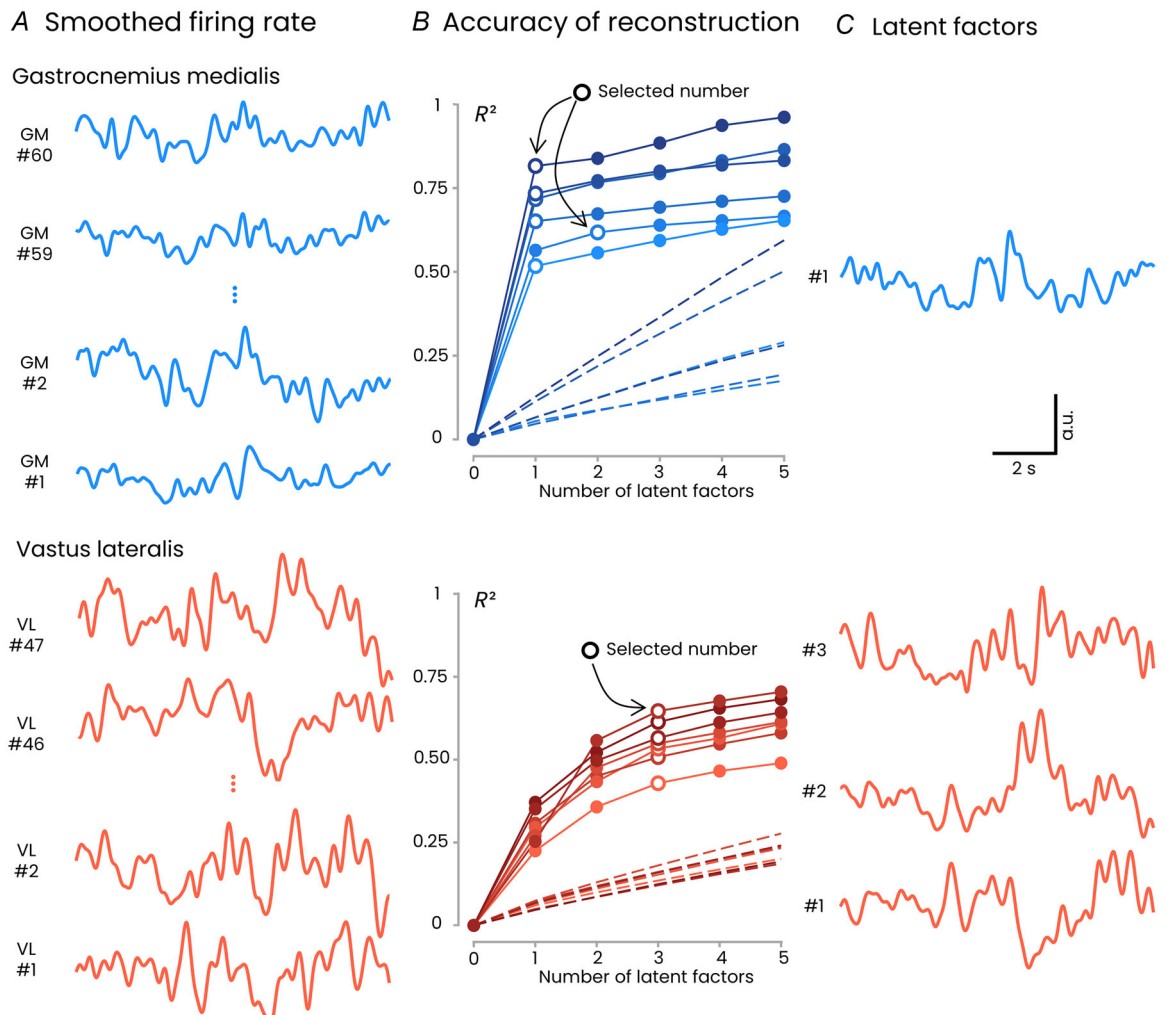

**Figure 1. Dimensionality of motor unit behaviour**
*A*, individual example of the smoothed firing rate of motor units identified in the GM (gastrocnemius medialis, upper panel) and the VL (vastus lateralis, lower panel). In these individual participants, 60 motor units were identified in the GM and 47 in the VL. These smoothed firing rate patterns were used as input to the factor analysis. *B*, for each number of latent factors (1–10, only five of which are shown here), we calculated the reconstruction accuracy as the $R^2$ between the reconstructed and the original data. Each participant is represented by a different shade of blue/red. The dashed lines represent the variance explained for the surrogate data generated by shuffling the inter-spike intervals of the recorded motor units (see Methods). The number of identified latent factors (white-filled dots) corresponds to the point at which the slope of the $R^2$ curve from experimental data becomes lower than the slope for the surrogate data. *C*, latent factors identified for an individual participant. a.u., arbitrary units. [Colour figure can be viewed at wileyonlinelibrary.com]

From a relatively low number of units identified in the RF (5.4 ± 4.6 motor units per participant), we identified higher flexibility for pairs involving RF compared to the VL–VL or GM–GM pairs (dispersion: +20% and +6% for RF–VL compared to VL–VL and GM–GM across the three conditions, respectively; displacement: +32% and +19% for RF–VL compared to VL–VL and GM–GM across the three conditions, respectively). This suggests that the flexibility observed in both the VL and GM motor units was comparatively small, even though the factor analysis identified three latent factors for the VL.

### Experiment 2: volitional control of motor units

We performed a second experiment to determine whether the distinct behaviours of the VL and GM motor units reflected a distinct dimensionality of their *volitional* control. Specifically, we assessed whether the participants were able to volitionally dissociate the activity of two motor units from either the GM or the VL when presented with online visual feedback of their firing rates. Visual feedback involved a cursor navigating in a 2D space, where the cursor coordinates corresponded to the firing rates of the two motor units, each identified in a different muscle region (proximal and distal, see Methods). Two target areas along the *x*- (Target 1) or *y*-axes (Target 2) were displayed to the participants (Fig. 3). To position the cursor within these target areas, the participants had to modulate the firing rates of the two motor units independently, namely increase the firing rate of one unit while maintaining or decreasing the firing rate of the other unit. We tested an average of 2.6 ± 0.5 (VL) and 1.8 ± 0.8 (GM) pairs of motor units per participant (a total of 13 pairs for VL and 9 pairs for GM). For each pair of motor units, the participants performed two sets of contractions at either 10% (VL) or 20% MVC (GM). Participants were instructed to reach Target 2

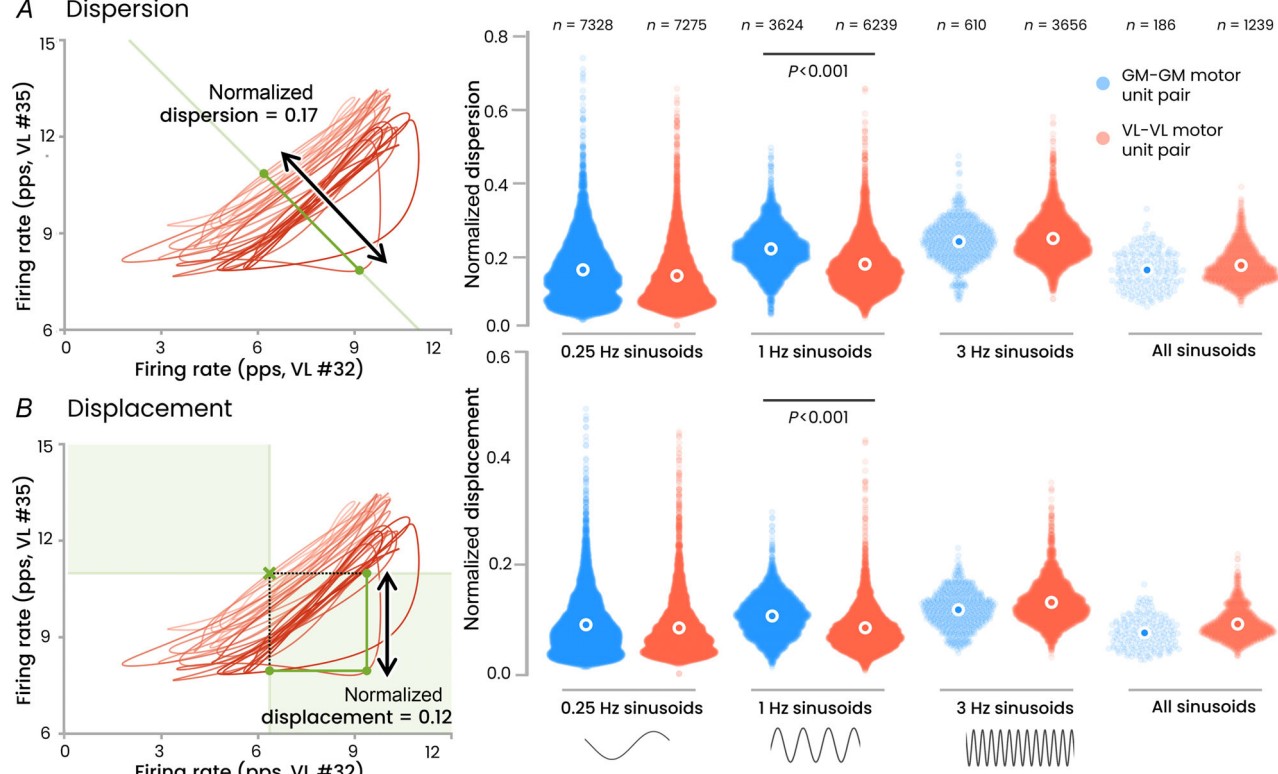

**Figure 2. Motor unit recruitment flexibility during sinusoidal tasks**
We evaluated the flexibility in motor unit recruitment during isometric contractions with sinusoidal force profiles at various frequencies (0.25, 1 and 3 Hz) and across all three frequencies combined using motor units matched across conditions ('all sinusoids'). Flexibility was quantified using two metrics: *dispersion* and *displacement* (Marshall et al., 2022). These metrics are based on the premise that the rigid control of motor units would constrain the trajectory of their firing rate along a one-dimensional monotonic manifold (see Methods). The right panels display the normalized values of the dispersion and displacement for each pair of GM–GM and VL–VL motor units, with sample sizes (*n*) indicated above. White circles represent the mean value. There was no difference between muscles, except for the 1 Hz condition, where GM–GM pairs exhibited slightly higher flexibility than VL–VL pairs. GM, gastrocnemius medialis; VL, vastus lateralis; pps, pulses per second. [Colour figure can be viewed at wileyonlinelibrary.com]

during the first set and Target 1 during the second set. The success rate was calculated as the percentage of time for which the participants maintained the cursor within the targets while producing the required torque (with a margin of ±5%) over the total duration of the contraction. For the purposes of analysis, the firing rate of the motor unit with the highest recruitment threshold was depicted on the *y*-coordinate (Target 2). Reaching Target 2 suggests that the participants successfully isolated the activation of the highest threshold unit, demonstrating flexible control. In contrast, reaching Target 1 could be achieved by reducing the neural drive, which could theoretically be compensated for by a synergist muscle, and therefore not necessarily indicate flexible control. The success rate associated with Target 1 was 30.4 ± 32.4% for VL and 27.1 ± 23.9% for GM. Interestingly, none of the participants was able to reach Target 2, regardless of the muscle (success rate = 0%).

We further analysed the behaviour of the motor units that were not displayed as feedback to the participants (15.6 ± 7.1 motor units per participant for the VL and 16.6 ± 6.2 motor units per participant for the GM). In very rare cases, we found that Target 2 was reached with these motor units, but with no difference between the two muscles (3.0 ± 12.8% of total time for VL and 3.4 ± 14.0% of total time for GM, *P* = 0.41, *t* test). Together, these results support a lack of flexibility in the volitional control of the motor units of the GM or VL muscles, despite the distinct dimensionality of their manifolds. This conclusion contrasts with the expectation that a greater dimensionality would correspond to a greater number of common inputs of cortical origin (Marshall et al., 2022). Therefore, we analysed other mechanisms that might explain the experimental observation of rigid control with the presence of multiple latent factors in the VL.

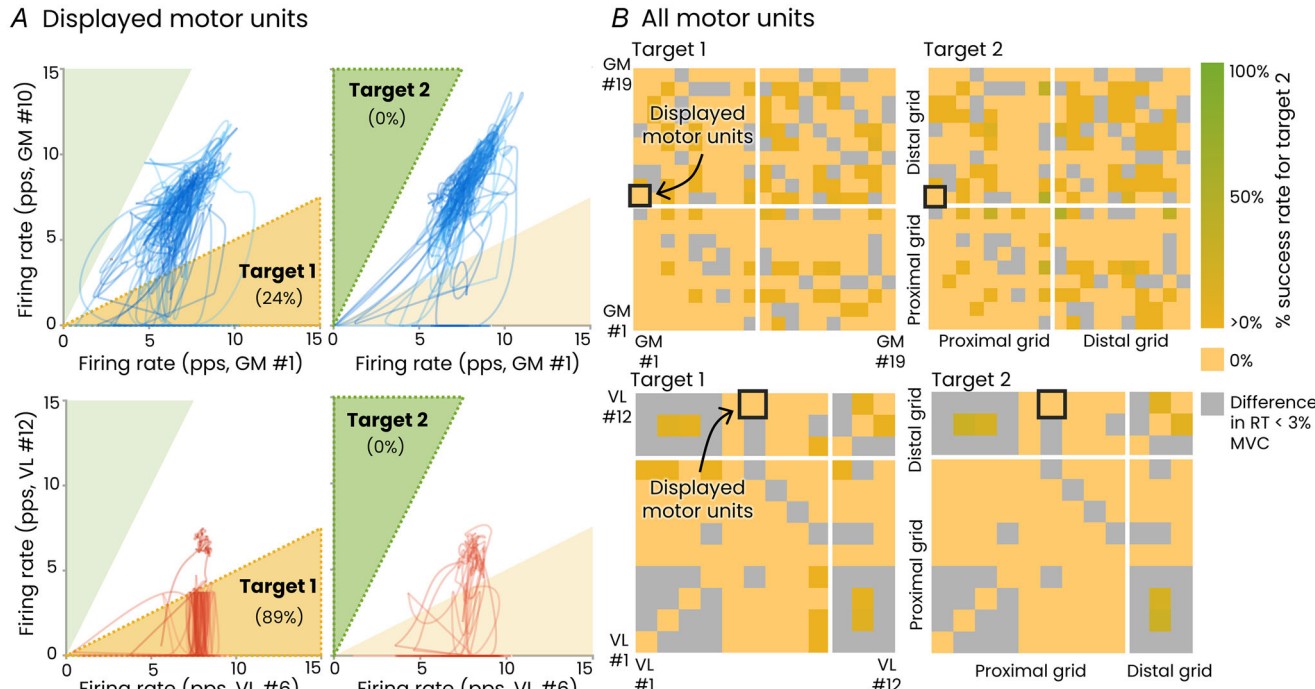

**Figure 3. The online control task with visual feedback**
This figure illustrates the results obtained for one representative participant for the gastrocnemius medialis (GM, top panels) and the vastus lateralis (VL, bottom panels). *A*, the visual feedback involved a cursor navigating in a 2D space, where the cursor coordinates corresponded to the firing rates of two motor units. Two target areas were displayed: one along the *x*-axis (Target 1) and one along the *y*-axis (Target 2). The dark shaded area represents the target the participant was instructed to reach, with the success rate (% of time spent in this target) indicated in parentheses. Note that the firing rate of the motor unit with the highest recruitment threshold is depicted on the *y*-axis (Target 2). Flexible control, defined as deviations from the size principle, is demonstrated by reaching Target 2, which was not achieved by this participant. *B*, for all the other pairs of motor units identified during this task, but not depicted as feedback, we calculated the success rate as the percentage of time during which the higher threshold motor units had a firing rate twice as high as the lower threshold motor units. This condition is equivalent to reaching Target 2. This analysis was only performed for pairs of motor units with a difference in recruitment threshold >3% of maximal voluntary contraction (MVC). The pair indicated as 'displayed motor units' are the pairs provided as feedback on *A*. The vast majority of motor units showed a success rate of 0%. RT, recruitment threshold; pps, pulses per second. [Colour figure can be viewed at wileyonlinelibrary.com]

## *In silico* models with spiking neural networks

We developed *in silico* models to explore the control mechanisms that might explain the differences observed between the GM and VL. Specifically, we tested scenarios of input distribution compatible with the multidimensionality of the linear manifold underlying the behaviour of VL motor units (Fig. 4). The seven scenarios are described in the Methods section. Factor analysis was applied to the activity of the *in silico* motor neurons to estimate the dimensionality of their linear manifolds, following the same approach used for the experimental data.

In the first scenario, which involved a homogeneous distribution of a single common input across motor neurons, only one latent factor was identified, confirming that a pool of motor neurons acts approximately linearly to transmit common inputs (Farina et al., 2014) Fig. 4*A* (Negro et al., 2009). Similarly, when three excitatory inputs (scenario ii, Fig. 4*B*) or three inhibitory inputs (scenario iv, Fig. 4*C*) were homogeneously distributed across motor neurons, a single latent factor was also identified. Although multiple inputs were present, their uniform distribution made them functionally equivalent to a single common input, albeit with the identified latent factor exhibiting a lower explained variance ($R^2$). Scenario vi, which involved homonymous recurrent inhibition, similarly resulted in the identification of a single latent factor, again with a lower explained variance (Fig. 4*D*). In contrast, when multiple inputs, either excitatory (scenario iii, Fig. 4*B*) or inhibitory (scenario v, Fig. 4*C*), were heterogeneously distributed across motor neurons, multiple latent factors were identified. Interestingly, a similar result was observed in scenario vii, which involved a single excitatory input combined with both homonymous and heteronymous recurrent inhibition (Fig. 4*E*). In that scenario, each motor neuron established excitatory connections exclusively with its homonymous Renshaw cells, and any heteronymous inhibitory input came from a separate, non-overlapping pool of Renshaw cells, thereby producing a multidimensional structure in the motor neuron activity.

It is important to note that the number of factors (three in scenarios iii, v and vii) was influenced by our chosen parameters, such as the number of input sources and the number of simulated pools. However, the key finding is not the specific number of latent factors but rather that the dimensionality of the motor unit manifold is mainly determined by how the inputs are distributed across the pool. Specifically, when multiple sources of input (excitatory or inhibitory) are distributed uniformly across all motor units, the motor unit activity remains confined within a 1D manifold. In contrast, when these inputs are distributed heterogeneously across motor units, additional dimensions emerge. Together, this demonstrates that it is the non-uniform projection of multiple inputs onto the motor units, rather than the mere presence of multiple inputs (excitatory or inhibitory), that increases the dimensionality of the manifold. Therefore, the differences in dimensionality observed experimentally between VL and GM probably arise from differences in the distribution of inputs and/or the configuration of recurrent inhibitory circuits.

## Discussion

We investigated the low-dimensional structure of the motor unit activity in two lower limb muscles. By applying a linear dimensionality reduction approach applied to a large sample of motor units, we found that the activity of the GM motor units was effectively captured by a single latent factor defining a unidimensional manifold, whereas the VL motor units were better represented by three latent factors defining a multidimensional manifold. Despite this apparent difference in the manifold dimensionality, the recruitment of motor units in the two muscles exhibited similarly limited flexibility. Using a spiking network model, we demonstrated that the multidimensional manifold formed by the VL motor unit firing activity is compatible with the presence of a single cortical command modulated by either non-homogeneously distributed inhibitory inputs or specific configurations of recurrent inhibitory circuits. Taken together, these results suggest that the dimensionality of motor unit linear manifolds is shaped by inputs from both cortical and subcortical sources and does not directly correspond to the dimensionality of volitional control.

There is evidence from anatomically compartmentalized muscles that synchronization between motor units is stronger within muscle compartments than between compartments (Keen & Fuglevand, 2004; McIsaac & Fuglevand, 2007; Reilly et al., 2004). Recent studies have further extended this observation to non-compartmentalized muscles, showing that the behaviour of motor units within the same pool may be more accurately represented by multidimensional linear manifolds (Del Vecchio et al., 2023; Hug et al., 2023; Madarshahian & Latash, 2021; Ricotta et al., 2023). These results align well with our finding that three latent factors are required to explain the variance in VL motor unit activity. As supported by our simulation (Fig. 4*B*), this could theoretically be explained by the presence of multiple independent supraspinal drives projecting to a single pool (Hug et al., 2023; Marshall et al., 2022). However, our experimental observations do not support this hypothesis. Specifically, we found that none of the participants was able to volitionally dissociate the activity of pairs of VL motor units. Furthermore, despite previous studies indicating that tasks with rapid changes in force

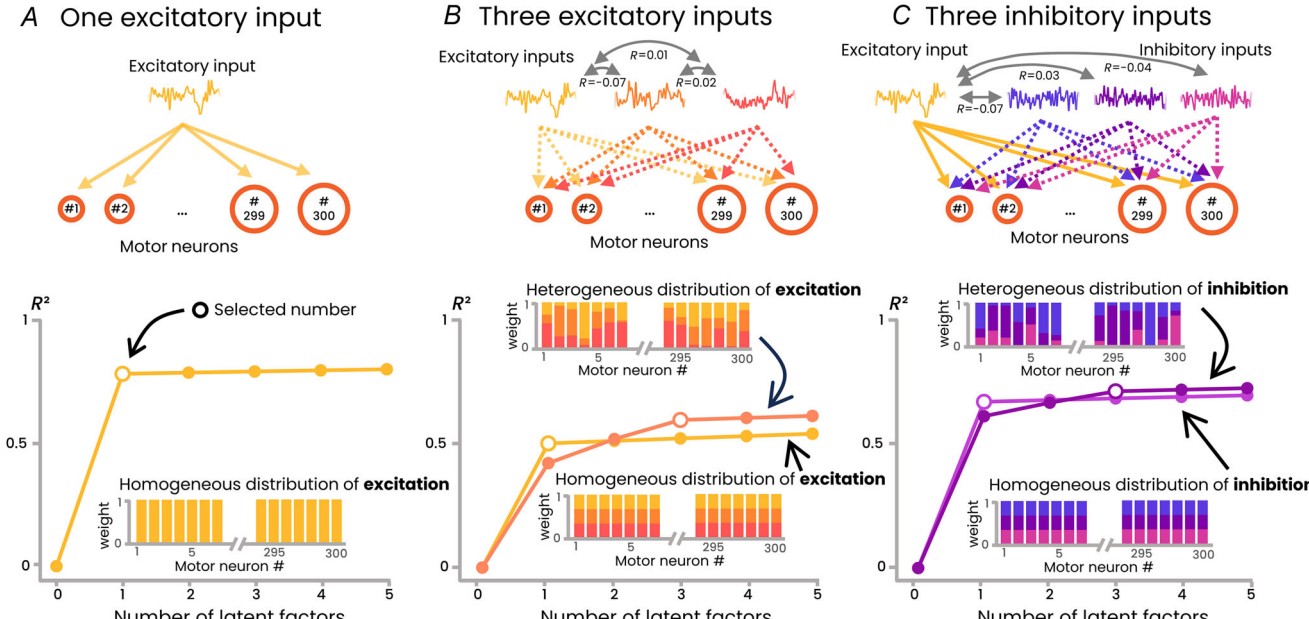

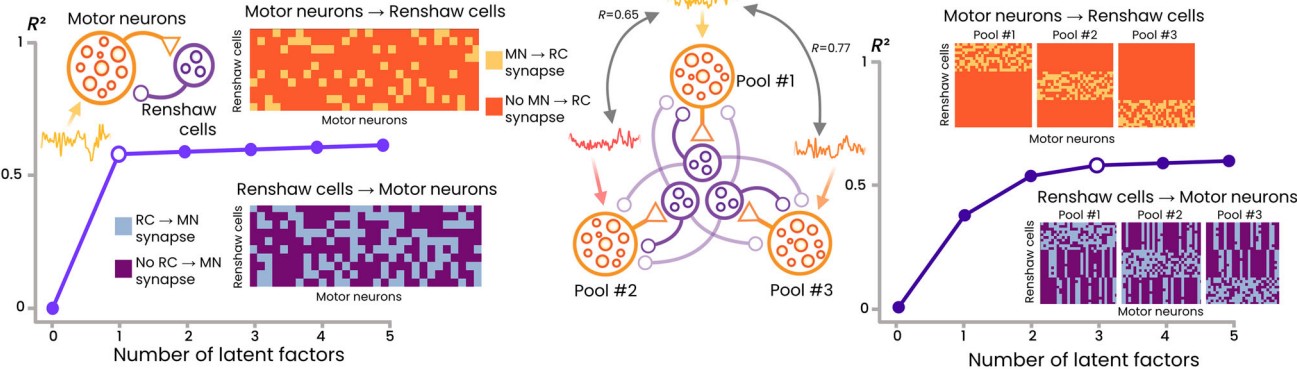

**Figure 4. Simulation scenarios**

*A*, a single common excitatory input (yellow) was homogenously distributed across 300 motor neurons (bar plot inset), resulting in the identification of one latent factor (scenario i). *B*, three uncorrelated excitatory inputs were distributed either homogeneously (scenario ii) or heterogeneously (scenario iii) to the motor neurons. An increase in the number of latent factors was observed only for the heterogeneous distributions. *C*, similar to *B*, but with three inhibitory inputs in addition to a uniformly distributed common excitatory input. As with excitatory inputs, a homogeneous distribution of inhibitory inputs (scenario iv) yielded a single latent factor, whereas a heterogeneous distribution resulted in three latent factors (scenario v). *D*, recurrent inhibition with random, homonymous connectivity (matrix insets, scenario vi). Providing recurrent inhibition to a single pool of motor neurons via Renshaw cells did not increase dimensionality. *E*, in addition to homonymous connectivity, recurrent inhibition was simulated with heteronymous connectivity between distinct motor neuron pools receiving correlated inputs, potentially representing synergistic muscles. Each pool had its own set of Renshaw cells. Importantly, these connections were non-overlapping across heteronymous pools, meaning that each motor neuron provided excitatory inputs exclusively to its own Renshaw cells, and any heteronymous inhibitory input originated from a unique Renshaw cell pool. This arrangement yielded multiple latent factors, demonstrating that non-uniform connectivity patterns can generate multidimensional manifolds despite the presence of a single common excitatory input. [Colour figure can be viewed at wileyonlinelibrary.com]

can trigger flexibility in motor unit recruitment (Grimby & Hannerz, 1977; Marshall et al., 2022), we did not observe any higher flexibility in the VL than in the GM during the fast oscillation tasks. Together, these findings indicate that multidimensionality in motor unit firing manifolds does not necessarily imply the presence of multiple cortical inputs and is therefore not a sufficient condition for flexible control. Interestingly, although several studies have previously reported multidimensional manifolds of the firings of motor units within (Ricotta et al., 2023) and across muscles (Rossato et al., 2024b), none has demonstrated clear flexible control in humans under constrained conditions (Heckman & Enoka, 2012). If the difference in dimensionality in the manifolds for the VL and GM is not due to a different number of cortical inputs, as suggested by the lack of flexibility in control, then a question arises as to how to explain this difference in dimensionality.

Spinal circuits involving muscle or tendon afferents (groups Ia, Ib, II, III and IV) are potential pathways that could contribute to increasing the dimensionality of the motor unit manifold, albeit with limited capacity for volitional control. For example, the distribution of Ia afferents to motor neurons within and between synergist muscles (Baldissera et al., 1981) suggests that Ia excitation could introduce additional common excitatory drives. Importantly, our simulation scenarios demonstrate that, regardless of the source of inputs from spinal circuits, these inputs must be distributed non-homogeneously across motor units to increase the dimensionality of the manifold (Fig. 4*B* and *C*). Of note, a heterogeneous distribution of inhibitory inputs has been suggested in response to nociceptive stimulation (Hodges et al., 2021; Hug et al., 2024). In addition, there is evidence of a regionalization of the stretch reflex in the VM muscle, where the activation of Ia afferents from a specific muscle region preferentially activates motor units within the same region (Gallina et al., 2017). However, in our study, we found no significant correlation between the distance separating two motor units on the skin surface and the degree of flexibility, which seems to rule out the involvement of this mechanism in our results. Of note, we are not aware of studies supporting differences in the organization of spinal circuits involving muscle or tendon afferents between the VL and GM muscles. Nonetheless, we cannot definitively exclude the potential contribution of these circuits to our findings.

Among the various neural pathways involved in the control of motor units, recurrent inhibition by Renshaw cells may play a crucial role in the behaviour observed in VL motor units. The role of Renshaw cells in decorrelating the activity of motor units within a pool has been previously discussed (Adam et al., 1978; Edgley et al., 2021; Maltenfort et al., 1998; Williams & Baker, 2009). Our simulation results indicate that the dynamics of the VL motor units are compatible with specific configurations of recurrent inhibitory circuits involving heteronymous connectivity, even with a single descending command. Specifically, our simulation aligned with the experimental data only when the model incorporated both homonymous and heteronymous recurrent inhibition (Fig. 4*E*). This was in the form of distinct inhibitory inputs driven by synergist muscles, which in our case were putatively the VM and vastus intermedius muscles. This finding is consistent with previous observations showing that Renshaw cells can be activated by recurrent collaterals of synergist motor neurons (Eccles et al., 1961; Edgley et al., 2021; Katz & Pierrot-Deseilligny, 1999). Importantly, the fact that homonymous recurrent inhibition alone did not increase the manifold dimensionality (Fig. 4*D*) suggests that it is not the amount of recurrent inhibition that shapes the manifold, but rather the connectivity of the recurrent inhibitory circuits in the spinal cord. Although we cannot exclude the possibility that the multidimensionality of the VL motor unit activity is an epiphenomenon arising from the complex circuitry of motor pathways serving no functional role, it seems improbable that biological systems would evolve sub-optimal muscle control schemes. Therefore, we argue that the distinct control schemes of the GM and VL are designed to comply with specific functional roles.

Considering the proposed role of recurrent inhibition in limiting the synchrony of motor unit firings (Maltenfort et al., 1998; Williams & Baker, 2009), heteronymous projections between motor pools probably serve to desynchronize the firings of motor units from different muscles. This may be particularly important for muscles that share a high level of common drive, as is the case between the VL and VM (Avrillon et al., 2021; Laine et al., 2015). In such muscle groups, controlling the timing of motor unit discharges to facilitate smooth force production would require both homonymous and heteronymous recurrent inhibitory circuits. In contrast, for muscles that share minimal common drives, such as the GM and its synergists (Hug, del Vecchio et al., 2021; Levine et al., 2023), heteronymous circuits would be unnecessary. Together, this supports the functional role of heteronymous recurrent inhibition and the idea that it is large for motor nuclei with consistent activity patterns, whereas it is absent or moderate for motor nuclei that are activated independently (Trank et al., 1999).

Three key methodological considerations must be addressed. First, our results were obtained at low contraction intensities (up to 20% MVC), reflecting primarily the behaviour of the lowest-threshold motor units. Since our approach required relatively long contractions, higher intensities would probably have induced fatigue, thereby introducing a confounding factor. Of

note, investigating higher contraction intensities, where recurrent inhibition is expected to be lower (Hultborn & Pierrot-Deseilligny, 1979), could help elucidate the role of recurrent inhibitory circuits in shaping the dimensionality of the motor unit manifold. Second, we acknowledge that multiple neural pathways could contribute to the observed between-muscle differences in manifold dimensionality. Although we used a spiking neural network model to explore possible scenarios, we cannot draw definitive conclusions about the specific neural circuits responsible for shaping this dimensionality. Third, a previous study involving a rhesus macaque performing isometric force-tracking tasks reported greater flexibility in motor unit recruitment with chirp force patterns compared to sinusoidal patterns (Marshall et al., 2022). In that study, the monkey was highly trained and capable of accurately tracking complex force patterns. In contrast, our human participants encountered significant challenges in following a 3 Hz sinusoidal force pattern. Introducing a chirp pattern would probably have further increased the difficulty of the task. For this reason, we opted to use a sinusoidal force profile at a constant frequency, balancing the need for dynamic force modulation with the participants' ability to perform the task accurately. Importantly, our conclusion regarding the lack of flexibility is not based solely on the sinusoidal tasks but is supported primarily by the online motor unit control tasks, where participants were free to explore any inputs to the motor units. This approach provided a more direct means for participants to demonstrate volitional, independent control; however, this was not observed. It is important to emphasize that these potential limitations do not undermine our primary conclusion that the multidimensionality of motor unit linear manifolds does not necessarily imply flexible control.

This study demonstrates that the activity of GM motor units can be effectively captured by a single latent factor defining a unidimensional manifold, whereas VL motor units are better represented by three latent factors defining a multidimensional manifold. Despite this difference in dimensionality, the recruitment of motor units exhibited similarly limited levels of flexibility. Our simulations revealed that a heterogeneous distribution of inputs to motor units, or specific configurations of spinal circuits, such as recurrent inhibition, could produce a multidimensional manifold, even with a single descending command. In conclusion, this study highlights that while motor unit firings can lie within a multidimensional manifold, the CNS may still lack the capacity for flexible volitional control over these motor units. By extension, these findings suggest that the dimensionality of the linear manifold may not directly reflect descending inputs but could instead be modulated by local spinal circuits.

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

## Additional information

### Data availability statement

The entire dataset (raw and processed data) and codes are publicly available at: https://doi.org/10.6084/m9.figshare.26364238.v1 and https://github.com/FrancoisDernoncourt/Motor_unit_flexibility.

### Competing interests

None declared.

### Author contributions

Contribution and design of the experiment: F.D., S.A. and F.H. Collection of data: F.D. Analysis and interpretation: F.D., S.A.,

T.L., T.C., D.F. and F.H. Drafting the article or revising it for important intellectual content: F.D., S.A., T.L., T.C., D.F. and F.H. All authors approved the final version of the manuscript.

## Funding

This study was funded by a grant from the French National Research Agency (ANR-24-CE17-5805; Neuromotor project). François Hug is supported by the French government, through the UCAJEDI Investments in the Future project managed by the ANR with reference number ANR-15-IDEX-01. Dario Farina is supported by the European Research Council Synergy Grant NaturalBionicS (contract #810346), the EPSRC Transformative Healthcare, NISNEM Technology (EP/T020970) and the BBSRC, 'Neural Commands for Fast Movements in the Primate Motor System' (NU-003743). The authors are grateful to the Université Côte d'Azur's Centre for High-Performance Computing (OPAL infrastructure) for providing resources and support.

## Acknowledgements

We thank Vincent Malejac (Université Jean-Monnet) for his assistance with data collection.

## Keywords

electromyography, factor analysis, motor neuron, muscle, synergy

## Supporting information

Additional supporting information can be found online in the Supporting Information section at the end of the HTML view of the article. Supporting information files available:

**Peer Review History**

