## [Peer Review History · The Journal of Physiology]

Flexible Control of Motor Units: Is the Multidimensionality of Motor Unit Manifolds a Sufficient Condition?

Francois Derroncourt, Simon Avrillon, Tijn Logtens, Thomas Cattagni, Dario Farina, and Francois Hug
DOI: 10.1113/JP287857

Corresponding author(s): Francois Hug (francois.hug@univ-cotedazur.fr)

Review Timeline:

Submission Date:	11-Oct-2024
Editorial Decision:	25-Nov-2024
Revision Received:	04-Jan-2025
Editorial Decision:	22-Jan-2025
Revision Received:	22-Jan-2025
Accepted:	27-Jan-2025

Senior Editor: Richard Carson

Reviewing Editor: Mathew Piasecki

Transaction Report:

Dear Dr Hug,

Re: JP-RP-2024-287857 "Flexible Control of Motor Units: Is the Multidimensionality of Motor Unit Manifolds a Sufficient Condition?" by Francois Dernoncourt, Simon Avrillon, Tijn Logtens, Thomas Cattagni, Dario Farina, and Francois Hug

Thank you for submitting your manuscript to The Journal of Physiology. It has been assessed by a Reviewing Editor and by 2 expert referees and we are pleased to tell you that it is potentially acceptable for publication following satisfactory major revision.

REVISION CHECKLIST:

Please upload two versions of your manuscript text: one with all relevant changes highlighted and one clean version with no

changes tracked. The manuscript file should include all tables and figure legends, but each figure/graph should be uploaded as separate, high-resolution files.

We look forward to receiving your revised submission.

Yours sincerely,

Richard Carson
Senior Editor
The Journal of Physiology

EDITOR COMMENTS

Reviewing Editor:

Comments to the Author:

Your paper has been assessed by two reviewers with expertise in this area. Both note a number of novel aspects of the study, however there are also a number of concerns that may limit its impact in current form, and these should be addressed before publication can be considered.

As raised by both reviewers, please pay particular attention to the limitations of i) the muscle groups chosen with regards to their functional role, ii) the relatively low contractions in which MUs were sampled. I appreciate this may be necessitated for the success of this methodological approach, but these limitations should be highlighted and discussed.

The authors are encouraged to be more convincing in their interpretation of the number of latent factors from sustained contractions in the VL, and to consider this outcome does not necessarily require additional excitatory or inhibitory inputs.

Senior Editor:

Comments to the Author:

As noted in comments provided by the referees and Reviewing Editor, it is essential that the scope of the modelling be extended to include the potential role of other elements of the spinal circuitry. You will discern that the referees have contributed several other valuable suggestions concerning ways in which the presentation and potential impact of your work might be enhanced. I would ask that you respond positively to these suggestions, such that the novel insights the work provides are transparent and convincing.

REFEREE COMMENTS

Referee #1:

This study addresses the extent of flexible motor unit (MU) control, a topic recently reported and debated in an invasive non-human primate study (Marshall et al., 2022). To this end, the authors utilized grid surface electromyography (sEMG) technology combined with a blind source separation (BSS) algorithm to decompose sEMG signals into constituent motor unit action potentials (MUAPs). Specifically, they aimed to identify the number of motor commands (referred to as common inputs or common drives) by performing factorization analysis on the discharges of decomposed MU populations in the thigh (Vastus Lateralis: VL) and lower leg (Medial Gastrocnemius: MG) muscles. This analysis was conducted on the sEMG data collected during a relatively simple ramp-and-hold (-down) isometric contraction task. Furthermore, the authors investigated whether human participants could independently control distinct MUs via an online feedback experiment.

Based on differences in the number of factors (presumably reflecting motor commands) observed between the two muscles

under identical tasks, the authors hypothesized that the number of factors derived from factorization does not solely represent descending cortical commands but is also influenced by local spinal circuitry. In particular, they focused on recurrent inhibition mediated by Renshaw cells in the spinal cord, which share axonal collaterals with spinal motoneurons innervating muscle fibers. To test this hypothesis, they developed a spiking neural network model. The results showed that specific configurations of recurrent inhibitory circuits could produce multiple factors, even with a single cortical command. The authors concluded that the number of factors may not correspond directly to descending inputs but instead be modulated by local spinal circuits. Additionally, the online feedback experiment demonstrated that participants could not control two MUs independently, leading the authors to conclude that MU control flexibility is limited.

Key Observations

- 1) Factorization analysis of MUs from surface multi-electrode recordings revealed either single or multiple latent factors during a simple ramp-and-hold contraction task, depending on the muscle.
- 2) Regardless of whether a muscle exhibited single or multiple factors, participants were unable to selectively increase the firing rate of high-threshold MUs without influencing already active low-threshold MUs.
- 3) The number of factors derived from factor analysis may not solely reflect the number of descending excitatory common inputs.

Critical Considerations

Although the findings are noteworthy, several concerns regarding the experimental design and analysis methods should be addressed:

- 1) The assumption of high flexibility in motor control for the thigh and lower leg muscles may not be well-supported, as these muscles are primarily adapted for generating large forces rather than fine motor control. While the authors justify their muscle selection based on potential differences in recurrent inhibition, the cited literature does not provide conclusive evidence supporting this claim. It is well-established that motor neurons innervating hand or paw muscles lack recurrent inhibition, but there is no clear evidence differentiating recurrent inhibition between thigh and lower leg muscles.
- 2) Marshall et al. reported MU firing patterns that deviated from conventional firing patterns in a chirp pattern, rather than in sinusoidal profiles. As such, a chirp task might have been more suitable for this investigation.
- 3) When assessing the dimensionality of control, comparing between tasks (rather than within a single task) may yield clearer insights. Rigid control theories predict consistency in motor commands across tasks, making such comparisons particularly valuable.
- 4) The proposal that recurrent inhibition explains the number of factors in VL raises questions about the specific contribution of this mechanism relative to other spinal feedback circuits. Spinal circuits involving Ia, Ib, group II afferents, or cutaneous feedback interact in complex, layered ways. Exploring the role of these alternative pathways through additional modules could provide further clarity.
- 5) The possibility that factorization introduces artifacts cannot be dismissed. Factorization inherently groups correlated activity, and the separation of factors observed during ramp-and-hold tasks may instead reflect distinct modes of activity at different frequencies. This could arise from groups of MUs firing at different rates based on recruitment thresholds, eliminating the need to invoke recurrent inhibition.
- 6) The use of sEMG raises the risk of signal contamination from neighboring muscles (crosstalk). Evaluating the spatial extent of signal spread is essential to ensure the validity of the findings.

7) The limited number of latent factors observed may inherently reflect the physiological properties of lower limb muscles, which are not specialized for fine motor control.

While the study presents intriguing findings, several methodological and theoretical concerns warrant further investigation to strengthen its conclusions. Addressing these points could provide deeper insights into the underlying mechanisms of MU control and the validity of factorization analysis in this context.

Referee #2:

This is an interesting study investigating how inputs to motoneurons are organised and whether there is any level of flexibility within the central nervous system to modulate the activity of independent motor units. I commend the authors for their comprehensive work, which includes experimental data and computer simulations, presented in this manuscript. I have some suggestions and considerations that I hope the authors will find helpful.

1 - An important point of consideration is the fact that contractions did not exceed 20% of MVC, and therefore these results reflect only the control of low-threshold motor units. Of note, both VL and GM recruit motor units at contraction forces above 90% of MVC, and therefore it is possible that these recruitment strategies and the contribution of different inputs could be different at contraction levels above 20%. Maybe the authors could consider a paragraph in the discussion section to elaborate their thoughts on what could happen at higher intensities and future research directions.

2 - Have the authors considered the influence of other inhibitory inputs present during the contraction. For example, autogenic inhibition might affect the amount of common synaptic input to motor units from the same muscle group. Autogenic inhibition is particularly greater at lower contraction levels in the gastrocnemius muscle group (see Khan and Burne, 2007).

Khan, S. I., & Burne, J. A. (2007). Reflex inhibition of normal cramp following electrical stimulation of the muscle tendon. *Journal of neurophysiology*, 98(3), 1102-1107.

3 - Given that the authors explored the effect of inhibition on decorrelating the activity of motor units, has any consideration been given to the effect of antagonist co-contractions (and therefore reciprocal inhibition) might have on common synaptic input? Has the activity of the antagonist group been measured in any of the experiments? I don't think this is necessarily an issue, but I was wondering whether the level of antagonist contractions might need to be considered in this type of experiment.

Line 179: Can you please justify the use of 10% of MVC for the knee extension and 20% for the plantarflexion task? Also, some consideration needs to be given to the different intensities when comparing results between different muscles.

Lines 601-604: The explanation about target 2 here makes sense, but how about target 1? What would reaching target 1 suggest? This hasn't been explained.

END OF COMMENTS

Point-by-point responses to the reviewers

We thank the reviewers and editors for their constructive feedback. In response to their comments, we have revised the manuscript and provided detailed responses to each point. We believe that these revisions have strengthened the manuscript.

Specifically, as suggested by the reviewers and editors, we expanded the scope of the modelling to include alternative organisations of excitatory and inhibitory inputs. To achieve this, **we updated the modelling approach** to allow independent adjustment of inhibitory and excitatory synaptic inputs, each with distinct conductance properties. While being more complex, this revised model is more appropriate for running new simulation scenarios. For transparency and reproducibility, the full code is **publicly available** at:

https://github.com/FrancoisDernoncourt/Motor_unit_flexibility.

Of note, as further detailed below, we would like to highlight to the editors and reviewers that the simulation component of this paper is (only) designed to discuss the main finding: *the presence of a multidimensional manifold does not necessarily imply the capacity for volitional and flexible control of motor units*.

Referee 1

This study addresses the extent of flexible motor unit (MU) control, a topic recently reported and debated in an invasive non-human primate study (Marshall et al., 2022). To this end, the authors utilized grid surface electromyography (sEMG) technology combined with a blind source separation (BSS) algorithm to decompose sEMG signals into constituent motor unit action potentials (MUAPs). Specifically, they aimed to identify the number of motor commands (referred to as common inputs or common drives) by performing factorization analysis on the discharges of decomposed MU populations in the thigh (Vastus Lateralis: VL) and lower leg (Medial Gastrocnemius: MG) muscles. This analysis was conducted on the sEMG data collected during a relatively simple ramp-and-hold (-down) isometric contraction task. Furthermore, the authors investigated whether human participants could independently control distinct MUs via an online feedback experiment.

Based on differences in the number of factors (presumably reflecting motor commands) observed between the two muscles under identical tasks, the authors hypothesized that the number of factors derived from factorization does not solely represent descending cortical commands but is also influenced by local spinal circuitry. In particular, they focused on recurrent inhibition mediated by Renshaw cells in the spinal cord, which share axonal collaterals with spinal motoneurons innervating muscle fibers. To test this hypothesis, they developed a spiking neural network model. The results showed that specific configurations of recurrent inhibitory circuits could produce multiple factors, even with a single cortical command. The authors concluded that the number of factors may not correspond directly to descending inputs but instead be modulated by local spinal circuits. Additionally, the online feedback experiment demonstrated that participants could not control two MUs independently, leading the authors to conclude that MU control flexibility is limited.

Key Observations

- 1) Factorization analysis of MUs from surface multi-electrode recordings revealed either single or multiple latent factors during a simple ramp-and-hold contraction task, depending on the muscle.
- 2) Regardless of whether a muscle exhibited single or multiple factors, participants were unable to selectively increase the firing rate of high-threshold MUs without influencing already active low-threshold MUs.
- 3) The number of factors derived from factor analysis may not solely reflect the number of descending excitatory common inputs.

Critical Considerations

Although the findings are noteworthy, several concerns regarding the experimental design and analysis methods should be addressed.

We thank the reviewer for their throughout review of our manuscript and their insightful comments, which guided changes that have strengthen the manuscript. These are detailed in the following.

Referee 1, comment 1

The assumption of high flexibility in motor control for the thigh and lower leg muscles may not be well-supported, as these muscles are primarily adapted for generating large forces rather than fine motor control. While the authors justify their muscle selection based on potential differences in recurrent inhibition, the cited literature does not provide conclusive evidence supporting this claim. It is well-established that motor neurons innervating hand or paw muscles lack recurrent inhibition, but there is no clear evidence differentiating recurrent inhibition between thigh and lower leg muscles.

We appreciate the opportunity to discuss this point further and we agree with the reviewer that our justification for muscle selection was not sufficiently supported. To address the comment on lack of evidence for differences in recurrent inhibition, we have now removed the following sentence from the last paragraph of the introduction: *“Moreover, crucially for muscle selection, the VL and GM likely have different levels of recurrent inhibition, as animal studies have shown that distal muscles generally exhibit less recurrent inhibition than proximal muscles”*.

Our study aimed to test the hypothesis that the presence of a multidimensional manifold does not necessarily imply the capacity for volitional, flexible control of motor units (see Introduction). Therefore, muscles were selected based on the expected dimensionality of their motor unit manifolds. Specifically, we selected a non-compartmentalized muscle in which the presence of a multidimensional manifold has been suggested from experimental evidence (VL; Del Vecchio et al., 2023, *J Neurosci*) and a second muscle from the same limb, known for exhibiting highly correlated motor unit activity (GM; Levine et al., 2023, *J Physiol*), as a ‘control.’ Thus, muscle selection is justified at the end of the introduction with the following rationale: *“To test this hypothesis, we recorded large samples of motor units to unravel the low-dimensional structures of their inputs. We selected two lower limb muscles based on previous studies reporting different levels of correlated activity of their motor units, with the vastus lateralis (VL) exhibiting less correlated activity (Avrillon et al., 2021) compared to the gastrocnemius medialis (GM) (Hug et al., 2021b)*

References:

- Avrillon S, Del Vecchio A, Farina D, Pons JL, Vogel C, Umehara J & Hug F (2021). Individual differences in the neural strategies to control the lateral and medial head of the quadriceps during a mechanically constrained task. *Journal of Applied Physiology* **130**, 269–281.
- Del Vecchio A, Marconi Germer C, Kinfé TM, Nuccio S, Hug F, Eskofier B, Farina D, Enoka RM. The Forces Generated by Agonist Muscles during Isometric Contractions Arise from Motor Unit Synergies. *J Neurosci*. 2023 Apr 19;43(16):2860-2873.
- Hug F, Del Vecchio A, Avrillon S, Farina D & Tucker K (2021*b*). Muscles from the same muscle group do not necessarily share common drive: evidence from the human triceps surae. *Journal of Applied Physiology* **130**, 342–354.
- Levine J, Avrillon S, Farina D, Hug F, Pons JL. Two motor neuron synergies, invariant across ankle joint angles, activate the triceps surae during plantarflexion. *J Physiol*. 2023 Oct;601(19):4337-4354.

Referee 1, comment 2

Marshall et al. reported MU firing patterns that deviated from conventional firing patterns in a chirp pattern, rather than in sinusoidal profiles. As such, a chirp task might have been more suitable for this investigation.

While Marshall et al. observed greater flexibility during chirp patterns, their study also reported flexibility during the sinusoidal contractions. The monkey in Marshall's study was highly trained and capable of precisely following complex force patterns. In contrast, our human participants encountered significant challenges in following a 3 Hz sinusoidal force pattern. Introducing a chirp pattern would likely have further increased the difficulty of the task. For this reason, we opted to use a sinusoidal force profile at a constant frequency, balancing the need for dynamic force modulation with the participants' ability to perform the task accurately. We believe this approach was sufficient to detect any flexibility present.

It is important to note that our conclusion regarding the lack of flexibility is not based solely on the sinusoidal tasks, but rather mainly stems from the online motor unit control paradigm, where participants were free to explore any control strategies that could potentially modify the inputs to the motor units. This approach provided a more direct means for participants to demonstrate volitional, independent control; however, this was not observed.

To address the reviewer's comment, the following text has been added in the Discussion section, in a new paragraph presenting the main limitations of our study: *“Third, a previous study involving a rhesus macaque performing isometric force-tracking tasks reported greater flexibility in motor unit recruitment with chirp force patterns compared to sinusoidal patterns. In that study, the monkey was highly trained and capable of accurately tracking complex force patterns. In contrast, our human participants encountered significant challenges in following a 3 Hz sinusoidal force pattern. Introducing a chirp pattern would likely have further increased the difficulty of the task. For this reason, we opted to use a sinusoidal force profile at a constant frequency, balancing the need for dynamic force modulation with the participants' ability to perform the task accurately. Importantly, our conclusion regarding the lack of flexibility is not based solely on the sinusoidal tasks but is primarily supported by the online motor unit control tasks, where participants were free to explore any inputs to the motor units. This approach provided a more direct means for participants to demonstrate volitional, independent control; however, this was not observed.”*

Referee 1, comment 3

When assessing the dimensionality of control, comparing between tasks (rather than within a single task) may yield clearer insights. Rigid control theories predict consistency in motor commands across tasks, making such comparisons particularly valuable.

The reviewer is correct. We initially calculated flexibility (dispersion and displacement) across the three sinusoidal tasks. However, this approach required tracking the same pairs of motor units across tasks, which significantly reduced the number of motor units included in the analysis. As a result, we opted to present flexibility results calculated for each task separately (i.e., 0.25, 1, and 3 Hz). Of note, whether flexibility was calculated from individual tasks or concatenated tasks, the results led to the same conclusion: flexibility does not show consistent differences between muscles.

To address the reviewer's comment, we included the analysis performed on all sinusoidal tasks in the revised version of the manuscript (Figure 1 below). It is noteworthy that the number of motor unit pairs decreases dramatically in this analysis, due to the challenge of tracking individual motor units across conditions.

Figure 1

The following sentence has been added to the Results section: “We also calculated dispersion and displacement across all three tasks concatenated, using motor units successfully tracked across all three tasks, resulting in 186 GM-GM pairs and 1239 VL-VL pairs.”

In addition, Fig. 2 in the manuscript has been updated with the figure provided above.

Referee 1, comment 4

The proposal that recurrent inhibition explains the number of factors in VL raises questions about the specific contribution of this mechanism relative to other spinal feedback circuits. Spinal circuits involving Ia, Ib, group II afferents, or cutaneous feedback interact in complex, layered ways. Exploring the role of these alternative pathways through additional modules could provide further clarity.

We appreciate the opportunity to further discuss this point. First, we would like to emphasize that the simulation component of this paper is (only) designed to discuss the main finding, that is, *the presence of a multidimensional manifold does not necessarily imply the capacity for volitional and flexible control of motor units*. However, we fully agree with the reviewer that the original version of the manuscript did not adequately acknowledge the potential contribution of other pathways.

To address the reviewer's comment, we updated the simulations to include additional scenarios with inhibitory inputs distributed either uniformly or heterogeneously across the motor units. This required revising the modelling approach to distinguish between excitatory and inhibitory inputs by adjusting membrane conductance relative to distinct inhibitory and excitatory potentials (please see the revised *Methods* and *Results* sections). The results suggest that the dimensionality of the motor unit manifold is mainly determined by how the excitatory and inhibitory inputs are distributed across the pool. Specifically, when multiple sources of input (excitatory or inhibitory) are distributed uniformly across all motor units, the motor unit activity remains confined within a one-dimensional manifold (Fig. 2 below). In contrast, when these inputs are distributed heterogeneously across motor units, additional dimensions emerged. Together, it demonstrates that it is the non-uniform projection of multiple inputs onto the motor units, rather than the mere presence of multiple inputs (excitatory or inhibitory), that increases the dimensionality of the manifold. Of note, even though we did not simulate specific pathways (such as those involving Ia, Ib, II, or cutaneous feedback), we believe this interpretation remains valid regardless of the pathways involved.

Figure 2. Simulation scenarios

[for review purpose only] Because reviewer 2 had a specific comment on autogenic inhibition, we implemented a simplified closed-loop scenario using Ib inhibitory interneurons. In this model, the collective force output of the motor unit pool was fed back through Ib interneurons, providing force-dependent inhibitory input to the motor neurons (Fig. 3 below). These results confirm that the dimensionality of the motor unit manifold arises from the distribution pattern of inhibitory inputs. Whether negative feedback is mediated by Renshaw or Ib interneurons, it is the nonuniform distribution of inputs—not the mechanism per se—that drives multidimensionality. For the sake of clarity, we decided not to include this specific scenario in the revised manuscript but have added further discussion about the potential pathways involved.

Figure 3. Autogenic inhibition [for review purpose only]

To address the reviewer's comment, we modified the discussion section to better acknowledge the potential contribution of other pathways: *“Spinal circuits involving muscle or tendon afferents (group Ia, Ib, II, III, and IV) are potential pathways that could contribute to increasing the dimensionality of the motor unit manifold, albeit with limited capacity for volitional control. For example, the distribution of Ia afferents to motor neurons within and between synergist muscles (Baldissera et al., 1981) suggests that Ia excitation could introduce additional common excitatory drives. Importantly, our simulation scenarios demonstrate that, regardless of the source of inputs from spinal circuits, these inputs must be distributed non-homogeneously across motor units to increase the dimensionality of the manifold (Figure 4B and C). Of note, a heterogenous distribution of inhibitory inputs has been suggested in response nociceptive stimulation (Hodges et al., 2021; Hug et al., 2024). In addition, there is evidence of a regionalization of the stretch reflex in the Vastus medialis muscle, where the activation of Ia afferents from a specific muscle region preferentially activates motor units within the same region (Gallina et al., 2017). However, in our study, we found no significant correlation between the distance separating two motor units on the skin surface and the degree of flexibility, which seems to rule out the involvement of this mechanism in our results. Of note, we are not aware of studies supporting differences in the organization of spinal circuits involving muscle or tendon afferents between the VL and GM muscles. Nonetheless, we cannot definitively exclude the potential contribution of these circuits to our findings.”*

We also updated the paragraphs discussing recurrent inhibition as follows: *“Among the various neural pathways involved in the control of motor units, recurrent inhibition by Renshaw cells may play a crucial role in the behaviour observed in VL motor units. The role of Renshaw cells in decorrelating the activity of motor units within a pool has been previously discussed (Adam et al., 1978; Maltenfort et al., 1998; Williams & Baker, 2009; Edgley et al., 2021). Our simulation results indicate that the dynamics of the VL motor units is compatible with specific configurations of recurrent inhibitory circuits involving heteronymous connectivity, even with a single descending command. Specifically, our simulation aligned with the experimental data only when the model incorporated both homonymous and heteronymous recurrent inhibition (Figure 4E). This was in the form of distinct inhibitory inputs driven by synergist muscles, which in our case were putatively the vastus medialis and vastus intermedius muscles. This finding is consistent with previous observations showing that Renshaw cells can be activated by recurrent collaterals of synergist motor neurons (Eccles et al., 1961; Katz & Pierrot-Deseilligny, 1999; Edgley et al., 2021). Importantly, the fact that homonymous recurrent inhibition alone did not increase the manifold dimensionality (Figure 4D) suggests that it is not the amount of recurrent inhibition that shapes the manifold, but rather the connectivity of the recurrent inhibitory circuits in the spinal cord. Although we cannot exclude the possibility that the multidimensionality of the VL motor unit activity is an epiphenomenon arising from the complex circuitry of motor pathways serving no functional role, it seems improbable that biological systems would evolve sub-optimal muscle control schemes. Therefore, we argue that the distinct control schemes of the GM and VL are designed to comply with specific functional roles.*

Considering the proposed role of recurrent inhibition in limiting the synchrony of motor unit firings (Maltenfort et al., 1998; Williams & Baker, 2009), heteronymous projections between motor pools likely serve to desynchronise the firings of motor units from different muscles. This may be particularly important for muscles that share a high level of common drive, as is the case between the VL and vastus medialis (Laine et al., 2015; Avrillon et al., 2021). In such muscle groups, controlling the timing of motor unit discharges to facilitate smooth force production would require both homonymous and heteronymous recurrent inhibitory circuits.

In contrast, for muscles that share minimal common drive, such as the GM and its synergists (Hug et al., 2021b; Levine et al., 2023), heteronymous circuits would be unnecessary. Together, it supports the functional role of heteronymous recurrent inhibition and the idea that it is large for motor nuclei with consistent activity patterns, whereas it is absent or moderate for motor nuclei that are activated independently (Trank et al., 1999).”

For the changes made to the Methods and Results sections, we kindly refer the reviewer to the track-changes version of the manuscript (Methods: Page 13-17, Lines 403-516; Results: Page 23-25, Lines 709-763).

References:

- Adam D, Windhorst U & Inbar GF (1978). The effects of recurrent inhibition on the cross-correlated firing patterns of motoneurons (and their relation to signal transmission in the spinal cord-muscle channel). *Biol Cybernetics* **29**, 229–235.
- Avrillon S, Del Vecchio A, Farina D, Pons JL, Vogel C, Umehara J & Hug F (2021). Individual differences in the neural strategies to control the lateral and medial head of the quadriceps during a mechanically constrained task. *Journal of Applied Physiology* **130**, 269–281.
- Baldissera F, Hultborn H & Illert M (1981). Integration in Spinal Neuronal Systems. In *Comprehensive Physiology*, 1st edn., ed. Terjung R, pp. 509–595. Wiley. Available at: <https://onlinelibrary.wiley.com/doi/10.1002/cphy.cp010212>
- Eccles JC, Eccles RM, Iggo A & Ito M (1961). Distribution of recurrent inhibition among motoneurons. *The Journal of Physiology* **159**, 479–499.
- Edgley SA, Williams ER & Baker SN (2021). Spatial and Temporal Arrangement of Recurrent Inhibition in the Primate Upper Limb. *J Neurosci* **41**, 1443–1454.
- Gallina A, Blouin J, Ivanova TD & Garland SJ (2017). Regionalization of the stretch reflex in the human vastus medialis. *J Physiol* **595**, 4991–5001.
- Hodges PW, Butler J, Tucker K, MacDonell CW, Poortvliet P, Schabrun S, Hug F & Garland SJ (2021). Non-uniform Effects of Nociceptive Stimulation to Motoneurons during Experimental Muscle Pain. *Neuroscience* **463**, 45–56.
- Hug F, Del Vecchio A, Avrillon S, Farina D & Tucker K (2021b). Muscles from the same muscle group do not necessarily share common drive: evidence from the human triceps surae. *Journal of Applied Physiology* **130**, 342–354.
- Hug F, Deroncourt F, Avrillon S, Thorstensen J, Besomi M, Hoorn W van den & Tucker K (2024). Heterogeneous distribution of inhibitory inputs among motor units as a key mechanism for motor adaptations to pain. 2024.10.05.616762. Available at: <https://www.biorxiv.org/content/10.1101/2024.10.05.616762v1>.
- Katz R & Pierrot-Deseilligny E (1999). Recurrent inhibition in humans. *Progress in Neurobiology* **57**, 325–355.
- Laine CM, Martinez-Valdes E, Falla D, Mayer F & Farina D (2015). Motor Neuron Pools of Synergistic Thigh Muscles Share Most of Their Synaptic Input. *Journal of Neuroscience* **35**, 12207–12216.
- Levine J, Avrillon S, Farina D, Hug F & Pons JL (2023). Two motor neuron synergies, invariant across ankle joint angles, activate the triceps surae during plantarflexion. *The Journal of Physiology* JP284503.
- Maltenfort MG, Heckman CJ & Rymer WZ (1998). Decorrelating Actions of Renshaw Interneurons on the Firing of Spinal Motoneurons Within a Motor Nucleus: A Simulation Study. *Journal of Neurophysiology* **80**, 309–323.

Trank TV, Turkin VV & Hamm TM (1999). Organization of recurrent inhibition and facilitation in motoneuron pools innervating dorsiflexors of the cat hindlimb. *Experimental Brain Research* 125, 344–352.

Williams ER & Baker SN (2009). Renshaw Cell Recurrent Inhibition Improves Physiological Tremor by Reducing Corticomuscular Coupling at 10 Hz. *J Neurosci* 29, 6616–6624.

Referee 1, comment 5

The possibility that factorization introduces artifacts cannot be dismissed. Factorization inherently groups correlated activity, and the separation of factors observed during ramp-and-hold tasks may instead reflect distinct modes of activity at different frequencies. This could arise from groups of MUs firing at different rates based on recruitment thresholds, eliminating the need to invoke recurrent inhibition.

While factor analysis is unlikely to be directly influenced by the mean discharge rate of motor units, we acknowledge that due to the non-linear input-output behaviour of motor neurons, the same input distributed across motor neurons with differing intrinsic properties could result in varying behaviours, potentially leading to the identification of multiple latent factors. However, it is unlikely that this explains the differences in manifold dimensionality observed between the VL and GM muscles, particularly at low contraction intensities, where the recruitment of low-threshold motor units with similar intrinsic properties is expected. We agree that this was important to verify. To address this, we conducted additional analyses to determine whether the representation of motor units in the latent space was influenced by their recruitment threshold or discharge rate:

- First, for each participant, we projected the motor units of the VL muscle onto the three identified latent factors. This process assigned each motor unit a position vector in the latent factor space, where the magnitude along each factor axis represented the correlation between the motor unit's smoothed discharge rates and the corresponding factor. Motor units were subsequently clustered based on their dominant factor, identified as the factor with the highest magnitude (Fig. 4, panel A).
- Second, we used logistic regression mixed models to determine whether motor units' mean firing rates and recruitment thresholds (dependent variables) varied across clusters (predictor variable), including participant as a random effect [*model <= lmer(dependent_variable ~ cluster + (cluster | subject))*"]. The results revealed no significant association between clusters and recruitment thresholds (P=0.323) or mean firing rates (P=0.17).

These findings suggest that the representation of motor units in the latent space—and, by extension, the factors identified through factorization—does not simply reflect intrinsic motor unit properties. These results increase our confidence that the factor analysis captured the dimensionality of the neural input to motor units, rather than artifacts arising from their intrinsic properties.

The following text has been added in the Methods section: *“We also examined whether factor analysis identified latent factors based on the intrinsic properties of motor units. For each participant, we projected the motor units onto the latent factors, assigning each motor unit a position vector in the factor space. The coordinates of this vector reflected the correlation between the unit's smoothed discharge rates and each factor. Motor units were subsequently clustered based on their dominant factor, identified as the factor with the highest magnitude.*

This approach allowed us to evaluate whether motor unit clustering was associated with firing rate and/or recruitment threshold.”

The following text has been added in the Results section: *“We also examined whether the latent factor representation of VL motor units was influenced by firing rates or recruitment thresholds rather than by the dimensionality of their inputs. No significant association emerged between dominant factors and either recruitment thresholds ($P = 0.323$) or mean firing rates ($P = 0.17$), suggesting that the dimensional structure is not determined by these intrinsic properties.”*

Figure 4 [for review purpose only].

Referee 1, comment 6

The use of sEMG raises the risk of signal contamination from neighboring muscles (crosstalk). Evaluating the spatial extent of signal spread is essential to ensure the validity of the findings.

While we acknowledge that crosstalk inevitably affects EMG signals collected with surface electrodes, it is important to note that our conclusions are based on motor unit data, not interferential EMG signals. The low power of crosstalk signals makes it unlikely for them to be decomposed into motor unit activity. This is supported by:

- i) simulation studies demonstrating that only motor units close to the electrodes (i.e. superficial units) are accurately decomposed (Avrillon et al., 2024 J Electro Kinesiol),
- ii) experimental findings indicating that less than 1.5% of motor units identified by electrodes overlying the VL muscle originate from the neighbouring muscles, the rectus femoris in that case (Rossato et al., 2022 – J Neurophysiol).

We have not made any changes to the manuscript in response to this comment.

References:

- Avrillon S, Hug F, Baker SN, Gibbs C, Farina D. Tutorial on MUedit: An open-source software for identifying and analysing the discharge timing of motor units from electromyographic signals. *J Electromyogr Kinesiol.* 2024; 77:102886. doi: 10.1016/j.jelekin.2024.102886.
- Rossato J, Tucker K, Avrillon S, Lacourpaille L, Holobar A, Hug F. Less common synaptic input between muscles from the same group allows for more flexible coordination strategies during a fatiguing task. *J Neurophysiol.* 2022 ; 1;127(2):421-433. doi: 10.1152/jn.00453.2021.

Referee 1, comment 7

The limited number of latent factors observed may inherently reflect the physiological properties of lower limb muscles, which are not specialized for fine motor control.

We agree with the reviewer that non-compartmentalized lower limb muscles, such as the vastus lateralis (VL) and gastrocnemius medialis (GM), are predominantly involved in gross motor tasks. Traditionally, it is assumed that motor units from such non-compartmentalized muscles receive a large proportion of common inputs (De Luca and Erim, 1994), which constrain their firings to a one-dimensional manifold. Our interest in the VL and GM stems from the fact that they exhibit distinct levels of dimensionality in their motor unit behaviour. In other word, we do not believe that our results support a “limited number of latent factors” but instead support a difference in the number of latent factors between two lower limb muscles. We have not made any changes to the manuscript in response to this comment.

Reference:

- De Luca CJ, Erim Z. Common drive of motor units in regulation of muscle force. *Trends Neurosci.* 1994 Jul;17(7):299-305. doi: 10.1016/0166-2236(94)90064-7.

While the study presents intriguing findings, several methodological and theoretical concerns warrant further investigation to strengthen its conclusions. Addressing these points could provide deeper insights into the underlying mechanisms of MU control and the validity of factorization analysis in this context.

We thank again the reviewer for their thorough review and hope that our revisions meet their expectations.

Referee 2

This is an interesting study investigating how inputs to motoneurons are organised and whether there is any level of flexibility within the central nervous system to modulate the activity of independent motor units. I commend the authors for their comprehensive work, which includes experimental data and computer simulations, presented in this manuscript. I have some suggestions and considerations that I hope the authors will find helpful.

We thank the reviewer for their throughout review of our manuscript and their insightful comments, which guided changes that have strengthen the manuscript.

Referee 2, comment 1

An important point of consideration is the fact that contractions did not exceed 20% of MVC, and therefore these results reflect only the control of low-threshold motor units. Of note, both VL and GM recruit motor units at contraction forces above 90% of MVC, and therefore it is possible that these recruitment strategies and the contribution of different inputs could be different at contraction levels above 20%. Maybe the authors could consider a paragraph in the discussion section to elaborate their thoughts on what could happen at higher intensities and future research directions.

The reviewer is correct that our observations are limited to relatively low contraction intensities. Higher contraction intensities present two main challenges. First, the occurrence of fatigue may act as a cofounding factor. Second, motor unit yield significantly decreases for contraction intensities > 20% MVC (Avrillon et al., 2024 – eLife).

To address the reviewer comment, the following text was added in the revised version of the Discussion “*First, our results were obtained at low contraction intensities (up to 20% MVC), primarily reflecting the behaviour of the lowest-threshold motor units. Since our approach required relatively long contractions, higher intensities would likely have induced fatigue, thereby introducing a confounding factor. Of note, investigating higher contraction intensities, where recurrent inhibition is expected to be lower (Hultborn & Pierrot-Deseilligny, 1979), could help elucidate the role of recurrent inhibitory circuits in shaping the dimensionality of the motor unit manifold.*”

Reference:

Avrillon S, Hug F, Enoka RM, Caillet AHD, Farina D. The identification of extensive samples of motor units in human muscles reveals diverse effects of neuromodulatory inputs on the rate coding. *Elife*. 2024 9;13:RP97085. doi: 10.7554/eLife.97085.
Hultborn H & Pierrot-Deseilligny E (1979). Changes in recurrent inhibition during voluntary soleus contractions in man studied by an H-reflex technique. *J Physiol* **297**, 229–251.

Referee 2, comment 2

Have the authors considered the influence of other inhibitory inputs present during the contraction. For example, autogenic inhibition might affect the amount of common synaptic input to motor units from the same muscle group. Autogenic inhibition is particularly greater at lower contraction levels in the gastrocnemius muscle group (see Khan and Burne, 2007). Khan, S. I., & Burne, J. A. (2007). Reflex inhibition of normal cramp following electrical stimulation of the muscle tendon. *Journal of neurophysiology*, 98(3), 1102-1107.

We appreciate the opportunity to discuss this point further. To address this comment, we implemented a simplified closed-loop scenario using Ib inhibitory interneurons [for review purpose only]. In this model, the collective force output of the motor unit pool was fed back through Ib interneurons, providing force-dependent inhibitory input to the motor neurons (Figure 5 below). These results confirm that the dimensionality of the motor unit manifold arises from the distribution pattern of inhibitory inputs. Whether negative feedback is mediated by Renshaw or Ib interneurons, it is the nonuniform distribution of inputs—not the mechanism per se—that drives multidimensionality.

Figure 5. Autogenic inhibition [for review purpose only]

For the sake of clarity, we decided not to include this specific scenario in the revised manuscript but have added further discussion about the potential pathways involved. Specifically, we updated the simulations to include additional scenarios with inhibitory inputs distributed either uniformly or heterogeneously across the motor units. This required revising the modelling approach to distinguish between excitatory and inhibitory inputs by adjusting membrane conductance relative to distinct inhibitory and excitatory potentials (please see the revised *Methods* and *Results* sections). The results suggest that the dimensionality of the motor unit manifold is mainly determined by how the excitatory and inhibitory inputs are distributed across the pool. Specifically, when multiple sources of input (excitatory or inhibitory) are distributed uniformly across all motor units, the motor unit activity remains confined within a one-dimensional manifold (Figure 2 in the responses to reviewer 1). In contrast, when these inputs are distributed heterogeneously across motor units, additional dimensions emerged. Together, it demonstrates that it is the non-uniform projection of multiple inputs onto the motor units, rather than the mere presence of multiple inputs (excitatory or inhibitory), that increases the dimensionality of the manifold. Of note, even though we did not simulate specific pathways (such as those involving Ia, Ib, II, or cutaneous feedback), we believe this interpretation remains valid regardless of the pathways involved.

To address the reviewer’s comment, we modified the discussion section to better acknowledge the potential contribution of other pathways: “*Spinal circuits involving muscle or tendon afferents (group Ia, Ib, II, III, and IV) are potential pathways that could contribute to increasing the dimensionality of the motor unit manifold, albeit with limited capacity for volitional control. For example, the distribution of Ia afferents to motor neurons within and between synergist muscles (Baldissera et al., 1981) suggests that Ia excitation could introduce additional common excitatory drives. Importantly, our simulation scenarios demonstrate that, regardless of the source of inputs from spinal circuits, these inputs must be distributed non-homogeneously across motor units to increase the dimensionality of the manifold (Figure 4B and C). Of note, a heterogenous distribution of inhibitory inputs has been suggested in response nociceptive stimulation (Hodges et al., 2021; Hug et al., 2024). In*

addition, there is evidence of a regionalization of the stretch reflex in the Vastus medialis muscle, where the activation of Ia afferents from a specific muscle region preferentially activates motor units within the same region (Gallina et al., 2017). However, in our study, we found no significant correlation between the distance separating two motor units on the skin surface and the degree of flexibility, which seems to rule out the involvement of this mechanism in our results. Of note, we are not aware of studies supporting differences in the organization of spinal circuits involving muscle or tendon afferents between the VL and GM muscles. Nonetheless, we cannot definitively exclude the potential contribution of these circuits to our findings.”

We also updated the paragraphs discussing recurrent inhibition as follows: *“Among the various neural pathways involved in the control of motor units, recurrent inhibition by Renshaw cells may play a crucial role in the behaviour observed in VL motor units. The role of Renshaw cells in decorrelating the activity of motor units within a pool has been previously discussed (Adam et al., 1978; Maltenfort et al., 1998; Williams & Baker, 2009; Edgley et al., 2021). Our simulation results indicate that the dynamics of the VL motor units is compatible with specific configurations of recurrent inhibitory circuits involving heteronymous connectivity, even with a single descending command. Specifically, our simulation aligned with the experimental data only when the model incorporated both homonymous and heteronymous recurrent inhibition (Figure 4E). This was in the form of distinct inhibitory inputs driven by synergist muscles, which in our case were putatively the vastus medialis and vastus intermedius muscles. This finding is consistent with previous observations showing that Renshaw cells can be activated by recurrent collaterals of synergist motor neurons (Eccles et al., 1961; Katz & Pierrot-Deseilligny, 1999; Edgley et al., 2021). Importantly, the fact that homonymous recurrent inhibition alone did not increase the manifold dimensionality (Figure 4D) suggests that it is not the amount of recurrent inhibition that shapes the manifold, but rather the connectivity of the recurrent inhibitory circuits in the spinal cord. Although we cannot exclude the possibility that the multidimensionality of the VL motor unit activity is an epiphenomenon arising from the complex circuitry of motor pathways serving no functional role, it seems improbable that biological systems would evolve sub-optimal muscle control schemes. Therefore, we argue that the distinct control schemes of the GM and VL are designed to comply with specific functional roles.*

Considering the proposed role of recurrent inhibition in limiting the synchrony of motor unit firings (Maltenfort et al., 1998; Williams & Baker, 2009), heteronymous projections between motor pools likely serve to desynchronise the firings of motor units from different muscles. This may be particularly important for muscles that share a high level of common drive, as is the case between the VL and vastus medialis (Laine et al., 2015; Avrillon et al., 2021). In such muscle groups, controlling the timing of motor unit discharges to facilitate smooth force production would require both homonymous and heteronymous recurrent inhibitory circuits. In contrast, for muscles that share minimal common drive, such as the GM and its synergists (Hug et al., 2021b; Levine et al., 2023), heteronymous circuits would be unnecessary. Together, it supports the functional role of heteronymous recurrent inhibition and the idea that it is large for motor nuclei with consistent activity patterns, whereas it is absent or moderate for motor nuclei that are activated independently (Trank et al., 1999).”

For the changes made to the Methods and Results sections, we kindly refer the reviewer to the track-changes version of the manuscript (Methods: Page 13-17, Lines 403-516; Results: Page 23-25, Lines 709-763).

References:

- Adam D, Windhorst U & Inbar GF (1978). The effects of recurrent inhibition on the cross-correlated firing patterns of motoneurons (and their relation to signal transmission in the spinal cord-muscle channel). *Biol Cybernetics* **29**, 229–235.
- Avrillon S, Del Vecchio A, Farina D, Pons JL, Vogel C, Umehara J & Hug F (2021). Individual differences in the neural strategies to control the lateral and medial head of the quadriceps during a mechanically constrained task. *Journal of Applied Physiology* **130**, 269–281.
- Baldissera F, Hultborn H & Illert M (1981). Integration in Spinal Neuronal Systems. In *Comprehensive Physiology*, 1st edn., ed. Terjung R, pp. 509–595. Wiley. Available at: <https://onlinelibrary.wiley.com/doi/10.1002/cphy.cp010212>
- Eccles JC, Eccles RM, Iggo A & Ito M (1961). Distribution of recurrent inhibition among motoneurons. *The Journal of Physiology* **159**, 479–499.
- Edgley SA, Williams ER & Baker SN (2021). Spatial and Temporal Arrangement of Recurrent Inhibition in the Primate Upper Limb. *J Neurosci* **41**, 1443–1454.
- Gallina A, Blouin J, Ivanova TD & Garland SJ (2017). Regionalization of the stretch reflex in the human vastus medialis. *J Physiol* **595**, 4991–5001.
- Hodges PW, Butler J, Tucker K, MacDonell CW, Poortvliet P, Schabrun S, Hug F & Garland SJ (2021). Non-uniform Effects of Nociceptive Stimulation to Motoneurons during Experimental Muscle Pain. *Neuroscience* **463**, 45–56.
- Hug F, Del Vecchio A, Avrillon S, Farina D & Tucker K (2021b). Muscles from the same muscle group do not necessarily share common drive: evidence from the human triceps surae. *Journal of Applied Physiology* **130**, 342–354.
- Hug F, Deroncourt F, Avrillon S, Thorstensen J, Besomi M, Hoorn W van den & Tucker K (2024). Heterogeneous distribution of inhibitory inputs among motor units as a key mechanism for motor adaptations to pain. 2024.10.05.616762. Available at: <https://www.biorxiv.org/content/10.1101/2024.10.05.616762v1>.
- Katz R & Pierrot-Deseilligny E (1999). Recurrent inhibition in humans. *Progress in Neurobiology* **57**, 325–355.
- Laine CM, Martinez-Valdes E, Falla D, Mayer F & Farina D (2015). Motor Neuron Pools of Synergistic Thigh Muscles Share Most of Their Synaptic Input. *Journal of Neuroscience* **35**, 12207–12216.
- Levine J, Avrillon S, Farina D, Hug F & Pons JL (2023). Two motor neuron synergies, invariant across ankle joint angles, activate the triceps surae during plantarflexion. *The Journal of Physiology* JP284503.
- Maltenfort MG, Heckman CJ & Rymer WZ (1998). Decorrelating Actions of Renshaw Interneurons on the Firing of Spinal Motoneurons Within a Motor Nucleus: A Simulation Study. *Journal of Neurophysiology* **80**, 309–323.
- Trank TV, Turkin VV & Hamm TM (1999). Organization of recurrent inhibition and facilitation in motoneuron pools innervating dorsiflexors of the cat hindlimb. *Experimental Brain Research* **125**, 344–352.
- Williams ER & Baker SN (2009). Renshaw Cell Recurrent Inhibition Improves Physiological Tremor by Reducing Corticomuscular Coupling at 10 Hz. *J Neurosci* **29**, 6616–6624.

Referee 2, comment 3

Given that the authors explored the effect of inhibition on decorrelating the activity of motor units, has any consideration been given to the effect of antagonist co-contractions (and

therefore reciprocal inhibition) might have on common synaptic input? Has the activity of the antagonist group been measured in any of the experiments? I don't think this is necessarily an issue, but I was wondering whether the level of antagonist contractions might need to be considered in this type of experiment.

This is an interesting comment. We did not measure the activity of the antagonist muscles, as it would have posed significant challenges for the hamstring muscles in the tested seated position. Based on previous work suggesting negligible coactivation during maximal knee extension (Avrillon et al., 2018), we do not expect a significant level of coactivation at this relatively low contraction intensity (20% MVC). However, we acknowledge that we lack data to confirm this claim.

Furthermore, to the best of our knowledge, it is unclear whether the same level of antagonist muscle activation produces an equivalent level of reciprocal inhibition in the quadriceps and calf muscles. This uncertainty would have complicated the interpretation of any information regarding the activity of the antagonist group.

We believe that the new scenarios tested in our simulations address this question, suggesting that it is not just the source of inhibitory inputs but their distribution pattern that shapes the dimensionality of the motor unit manifold. Therefore, for reciprocal inhibition to increase the manifold dimensionality, it would need to be heterogeneously distributed across the VL motor unit pool. Experimentally modulating the activity of antagonist muscles while maintaining constant activity in the agonist would provide an elegant approach to test the impact of reciprocal inhibition. Please refer to our response to comment 2 for details on the specific changes made in the revised version of the manuscript.

Reference:

Avrillon S, Hug F, Guilhem G. Between-muscle differences in coactivation assessed using elastography. *J Electromyogr Kinesiol.* 2018 43:88-94. doi: 10.1016/j.jelekin.2018.09.007.

Referee 2, comment 4

Line 179: Can you please justify the use of 10% of MVC for the knee extension and 20% for the plantarflexion task? Also, some consideration needs to be given to the different intensities when comparing results between different muscles.

The following sentence has been added to the revised version of the manuscript: “*The lower torque level for the knee extension tasks was chosen to minimize fatigue, which develops faster during knee extension than plantarflexion (Rossato et al., 2024a).*”

Reference:

J. Rossato, S. Avrillon, K. Tucker, D. Farina, F. Hug, The Volitional Control of Individual Motor Units Is Constrained within Low-Dimensional Neural Manifolds by Common Inputs. *J. Neurosci.* **44** (2024).

Referee 2, comment 5

Lines 601-604: The explanation about target 2 here makes sense, but how about target 1? What would reaching target 1 suggest? This hasn't been explained.

The text has been modified to read: “Reaching Target 2 suggests that the participants successfully isolated the activation of the highest threshold unit, demonstrating flexible control. In contrast, reaching Target 1 could be achieved by reducing the neural drive, which could theoretically be compensated by a synergist muscle, and therefore not necessarily indicate flexible control. The success rate associated with Target 1 was $30.4 \pm 32.4\%$ for VL and $27.1 \pm 23.9\%$ for GM. Interestingly, none of the participants were able to reach Target 2, regardless of the muscle (success rate = 0%).”

Dear Dr Hug,

Re: JP-RP-2025-287857R1 "Flexible Control of Motor Units: Is the Multidimensionality of Motor Unit Manifolds a Sufficient Condition?" by Francois DERNONCOURT, Simon AVRILLON, Tijn LOGTENS, Thomas CATTAGNI, Dario FARINA, and Francois HUG

Thank you for submitting your manuscript to The Journal of Physiology. It has been assessed by a Reviewing Editor and by 2 expert referees and we are pleased to tell you that it is acceptable for publication following satisfactory revision.

REVISION CHECKLIST:

We look forward to receiving your revised submission.

Yours sincerely,

Richard Carson
Senior Editor
The Journal of Physiology

EDITOR COMMENTS

Reviewing Editor:

Comments to the Author:

Thank you for your response which has been viewed largely favourably. However reviewer #1 has made an interesting point which necessitates further clarification on the method and its potential limitations.

REFEREE COMMENTS

Referee #1:

I appreciate the authors' efforts in addressing my previous comments, revisiting analyses, and revising the manuscript. Overall, the revisions have significantly clarified and strengthened the work. However, a few concerns remain that I believe should be addressed to further enhance the manuscript.

In response to my third comment, the authors have addressed flexibility across the three sinusoidal tasks by attempting to identify the same pairs of motor units across tasks. While the revisions provide valuable insights, I still find it challenging to directly compare differences in the flexibility of common inputs between single tasks and all sinusoidal tasks due to differences in the population sizes. It would be more appropriate to perform these comparisons using the same population of motor units, especially when analyzing dispersion and displacement metrics (e.g., 0.25 Hz vs. all, 1 Hz vs. all, 3 Hz vs. all).

The authors report a significantly reduced number of motor units identified consistently across tasks. While I find this result interesting, it also raises concerns regarding its implications for the authors' claim. Specifically, I am curious whether the reduction is primarily due to technical challenges in decomposing motor unit action potentials from surface EMG recordings. If so, could the authors elaborate on the nature of these challenges?

Alternatively, if the reduction is not solely due to technical issues, it may suggest that other motor units were recruited under different task conditions. This could indicate the deployment of distinct motor unit recruitment strategies, implying that different motor commands influenced the motoneuron pool. Such a finding might contradict the authors' claim of virtual single motor commands regulating the motoneuron pool, as suggested by the reported flexibility metrics.

Referee #2:

The authors have addressed all my comments adequately. I have no further comments.

END OF COMMENTS

Point-by-point responses to the reviewers

Referee #1:

I appreciate the authors' efforts in addressing my previous comments, revisiting analyses, and revising the manuscript. Overall, the revisions have significantly clarified and strengthened the work. However, a few concerns remain that I believe should be addressed to further enhance the manuscript.

We thank the reviewer for their throughout review of our manuscript.

1) In response to my third comment, the authors have addressed flexibility across the three sinusoidal tasks by attempting to identify the same pairs of motor units across tasks. While the revisions provide valuable insights, I still find it challenging to directly compare differences in the flexibility of common inputs between single tasks and all sinusoidal tasks due to differences in the population sizes. It would be more appropriate to perform these comparisons using the same population of motor units, especially when analyzing dispersion and displacement metrics (e.g., 0.25 Hz vs. all, 1 Hz vs. all, 3 Hz vs. all).

As mentioned in the manuscript, we chose not to report the contraction effect (i.e., the comparison between sinusoids at different frequencies), as these comparisons are not relevant to the aims of this study. Since our goal was to compare flexibility between muscles, we focused solely on between-muscle comparisons. In this context, the inability to track motor units across tasks (i.e., sinusoids at different frequencies) is not problematic, as it does not affect our conclusions.

For the reviewer's consideration, we performed the requested analysis but chose not to include it in the manuscript. When considering only the pairs of motor units that were matched across all sinusoidal contractions, we observed results consistent with those reported in the manuscript. Specifically, there was no significant main effect of Muscle for either dispersion ($P = 0.64$) or displacement ($P = 0.66$). However, there was a significant main effect of Contraction for both dispersion ($P = 0.002$) and displacement ($P < 0.001$), as well as a significant Muscle \times Contraction interaction ($P < 0.001$ for both metrics). Post-hoc pairwise comparisons for the contraction effect are displayed in Figure 1 at the end of this document (for review purposes only). Of note, the number of pairs differed drastically between muscles, i.e. 1239 pairs for VL vs. only 186 pairs for GM.

2) The authors report a significantly reduced number of motor units identified consistently across tasks. While I find this result interesting, it also raises concerns regarding its implications for the authors' claim. Specifically, I am curious whether the reduction is primarily due to technical challenges in decomposing motor unit action potentials from surface EMG recordings. If so, could the authors elaborate on the nature of these challenges? Alternatively, if the reduction is not solely due to technical issues, it may suggest that other motor units were recruited under different task conditions. This could indicate the deployment of distinct motor unit recruitment strategies, implying that different motor commands influenced the motoneuron pool. Such a finding might contradict the authors' claim of virtual single motor commands regulating the motoneuron pool, as suggested by the reported flexibility metrics.

We appreciate the opportunity to discuss this point further. The difficulty in tracking motor units across tasks likely arises from limitations of the decomposition approach. For example, tracking smaller motor units becomes challenging during contractions performed at higher intensities than those at which they were initially identified, due to the lower signal-to-noise

ratio when larger motor units are recruited. Consequently, when a motor unit cannot be tracked, it is impossible to distinguish between its non-recruitment and an inability to identify it.

Importantly, as indicated in our response to comment #1, this does not impact the conclusions of this article, which focuses on between-muscle comparisons.

To address the reviewer's comment, the following text has been added in the revised version of the manuscript: *"The difficulty in tracking motor units across contractions with differing mechanical constraints - and thus different activation levels (e.g. trapezoidal vs. sinusoidal tasks) - likely arises from technical limitations of the decomposition approach. Specifically, tracking smaller motor units becomes challenging during contractions performed at higher intensities than those at which they were initially identified, due to the reduced signal-to-noise ratio when larger motor units are recruited."*

Referee #2:

The authors have addressed all my comments adequately. I have no further comments.

We thank the reviewer for their throughout review of our manuscript.

Fig. 1. Normalised values of the dispersion and displacement for each pair of GM-GM and VL-VL motor units, with sample sizes (n) indicated above. White circles represent the mean value. Notably, the number of GM motor unit pairs used for the analysis for the 3 Hz sinusoidal contraction is lower (47 pairs). This is because we were unable to track GM motor units at this frequency in two participants. Consequently, these participants were not represented in the 3 Hz sinusoidal contraction but were included in the 'all sinusoids' condition with their 0.25 Hz and 1 Hz values.

Dear Professor Hug,

Re: JP-RP-2025-287857R2 "Flexible Control of Motor Units: Is the Multidimensionality of Motor Unit Manifolds a Sufficient Condition?" by Francois DERNONCOURT, Simon AVRILLON, Tijn LOGTENS, Thomas CATTAGNI, Dario FARINA, and Francois HUG

We are pleased to tell you that your paper has been accepted for publication in The Journal of Physiology.

Yours sincerely,

Richard Carson
Senior Editor
The Journal of Physiology

If you would like to receive our 'Research Roundup', a monthly newsletter highlighting the cutting-edge research published in The Physiological Society's family of journals (The Journal of Physiology, Experimental Physiology, Physiological Reports, The Journal of Nutritional Physiology and The Journal of Precision Medicine: Health and Disease), please click this link, fill in your name and email address and select 'Research Roundup':

<https://www.physoc.org/journals-and-media/membernews>

- **TRANSPARENT PEER REVIEW POLICY:** To improve the transparency of its peer review process, The Journal of Physiology publishes online as supporting information the peer review history of all articles accepted for publication. Readers will have access to decision letters, including Editors' comments and referee reports, for each version of the manuscript as well as any author responses to peer review comments. Referees can decide whether or not they wish to be named on the peer review history document.
- You can help your research get the attention it deserves! Check out Wiley's free Promotion Guide for best-practice recommendations for promoting your work at: www.wileyauthors.com/eeo/guide. You can learn more about Wiley Editing Services which offers professional video, design, and writing services to create shareable video abstracts, infographics, conference posters, lay summaries, and research news stories for your research at: www.wileyauthors.com/eeo/promotion.
- **IMPORTANT NOTICE ABOUT OPEN ACCESS:** To assist authors whose funding agencies mandate public access to published research findings sooner than 12 months after publication, The Journal of Physiology allows authors to pay an Open Access (OA) fee to have their papers made freely available immediately on publication.

EDITOR COMMENTS

Reviewing Editor:

Thank you for providing the additional analyses and clarification.

REFeree COMMENTS

Referee #1:

I think the authors addressed the points I raised, and I understood some limitations and caveats in the current study.